# QUANTITATIVE PERFORMANCE ASSESSMENT OF CNN UNITS VIA TOPOLOGICAL ENTROPY CALCULATION

**Yang Zhao & Hao Zhang**
Department of Electronic Engineering
Tsinghua University
`zhao-yan18@mails.tsinghua.edu.cn, haozhang@tsinghua.edu.cn`

## ABSTRACT

Identifying the status of individual network units is critical for understanding the mechanism of convolutional neural networks (CNNs). However, it is still challenging to reliably give a general indication of unit status, especially for units in different network models. To this end, we propose a novel method for quantitatively clarifying the status of single unit in CNN using algebraic topological tools. Unit status is indicated via the calculation of a defined topological-based entropy, called feature entropy, which measures the degree of chaos of the global spatial pattern hidden in the unit for a category. In this way, feature entropy could provide an accurate indication of status for units in different networks with diverse situations like weight-rescaling operation. Further, we show that feature entropy decreases as the layer goes deeper and shares almost simultaneous trend with loss during training. We show that by investigating the feature entropy of units on only training data, it could give discrimination between networks with different generalization ability from the view of the effectiveness of feature representations.

## 1 INTRODUCTION

Convolutional neural networks (CNNs) have achieved great success in various vision tasks (Szegedy et al., 2016; Redmon et al., 2016; He et al., 2017a). The key to such success is the powerful ability of feature representations to input images, where network units[1] play a critical role. But impacted by the diverse training deployments and huge hypothesis space, networks even with the same architecture may converge to different minima on a given task. Although units between these networks could present similar function for the same task, yet they may have completely different activation magnitudes. Consequently, this makes it fairly hard to give a general indication of the status for a given network unit with respect to how well features are represented by it from images in the same class.

Being rough indicators in practice, magnitude responses of units are usually chosen simply (Zhang et al., 2018) or processed statistically (such as average mean) (Li et al., 2016; Luo et al., 2017) based on the idea of matched filtering. However, firstly these indicators are apparently sensitive to rescaling operations in magnitude. If performing a simply rescaling operation to the weights such as the strategy introduced in Neyshabur et al. (2015), the results of the network and the function of each unit would all remain unchanged, but these indicators would vary along with the rescaling coefficient. Secondly, as the spatial information in the unit is completely discarded, they could not give discrimination between units with and without random patterns, for example units separately outputted by a well-trained and random-initialized CNN filter. Without a valid indication regarding the mentioned situations, these indicators fail to ensure the universal applicability for units in different network models.

In this paper, we attempt to investigate the status of units from a new perspective. Roughly speaking, natural images in the same class have common features, and meanwhile the locations of these features are spatially correlated in global. For effective units, features are picked out and represented

---

[1]Regarding the term unit, a unit is the perceptive node in networks, which generally refers to the activated feature map outputted by a convolutional filter in CNNs.

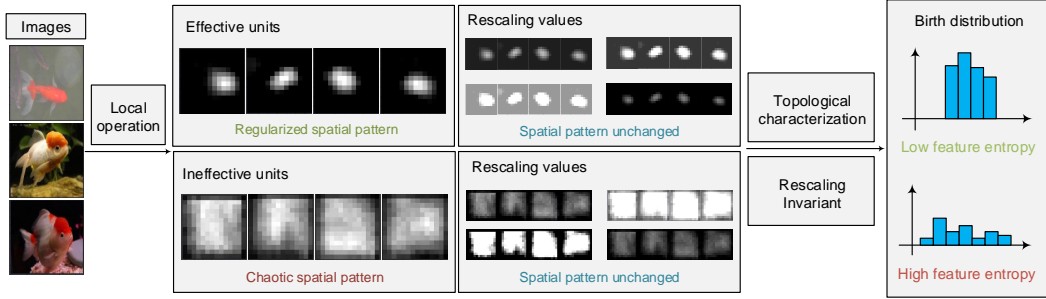

Figure 1: Comparisons between the effective units and ineffective units. For effective units, since the spatial pattern of the features in the images would be preserved, units should stably present this regularized spatial pattern. We propose a topological-based quantity called feature entropy to indicate the unit status, giving reliable indication in various situations like rescaling the values.

by high activation values in the units. And due to the locality nature in feature extraction by convolution, this global spatial pattern between the common features would be preserved synchronously in the counterpart representations in the effective units. In contrast, for ineffective units, being incapability of effectively representing these common features, representations would be in chaos and marks of this pattern is vague. This provides a valid road for performance assessment of individual units, and critically it is rescaling-invariant and universally applicable to any CNN architecture.

The investigation of such pattern could naturally lead to topological approaches because knowledge of topological data analysis such as barcodes (Ghrist, 2008) provides valuable tools to resolve the intrinsic patterns in raw data. Along this line, firstly we introduce a method for characterizing the spatial pattern of feature representations in units for a single sample by incorporating with the topological tools, and then use information entropy to evaluate the stability of this spatial characterizations for various images sampled from the same class, where we call it feature entropy. In this way, a unit is judged to be effective if its feature entropy is high, otherwise ineffective.

In our experiments, we find that feature entropy would gradually decrease as the layer goes deeper and the evolution trends of feature entropy and losses are almost the same during network training. We show that the feature entropy could provide reliable indication of unit status in situations like weight-rescaling and the emergence of random pattern. Finally, we show the value of feature entropy in giving discrimination between networks with different generalization ability by investigating only the training set.

## 2 RELATED WORKS

One line of research that attracts many researchers is seeking solutions in a way of visualizing what features have learned by the units (Zeiler & Fergus, 2014; Zhou et al., 2014; Mahendran & Vedaldi, 2015; Simonyan et al., 2013). Status is generally identified depending on the degree of alignment between the visualized features and the human-visual concepts (Bau et al., 2017; Zhou et al., 2018a; Bau et al., 2020). On the one hand, they meanwhile give excellent visual interpretation of each unit; on the other hand, it hinders its universal application to arbitrary tasks and models in which units' functionalities may be unrecognized to human (Wang et al., 2020).

Another related research trace lies in the field of network pruning, where they concentrate on using simple methods to roughly select less important units within a network. Typical approaches include the L1-Norm of units (Luo et al., 2017), Average Percentage of Zeros (APoZ) in units (Hu et al., 2016), some sparse-based methods (Li et al., 2019; Yoon & Hwang, 2017), and so on. Despite commonly used in practice, since without a specific processing on units in diverse situations, they are unable to provide a general indication for units in different networks.

Besides, Morcos et al. (2018) introduce the class selectivity from neuroscience to investigate the selectivity over classes for a specific unit, on the basis of calculating the mean units. Alain & Bengio (2016) propose linear classifier probe, where they report the degree of linear classification of units in intermediate layers could somehow characterize the status of units.

Lastly, we would like to discuss some recent works related to topological approaches in deep learning. Naitzat et al. (2020) demonstrate the superiority of using ReLu activation by studying the changes in Betti numbers of a two-class neural network. Montúfar et al. (2020) use neural networks to predict the persistent homology features. In Gabrielsson & Carlsson (2019), by using barcode, they show the topological structure changes during training which correlates to the generalization of networks. Rieck et al. (2018) propose the neural persistence, a topological complexity measure of network structure that could give a criterion on early stopping. Guss & Salakhutdinov (2018) empirically investigate the connection between neural network expressivity and the complexity of dataset in topology. In Hofer et al. (2017), topological signatures of data are evaluated and used to improve the classification of shapes.

## 3  METHOD

In general, input images for a network model are commonly resized to be square for processing. For input image sample $\boldsymbol{I}$ of a given class with size $n \times n$ to be represented by a unit $\boldsymbol{U}$ with size $m \times m$ via feature extraction processing $f$ in CNN, we have,

$$f : \boldsymbol{I} \to \boldsymbol{U} \tag{1}$$

For image $\boldsymbol{I}$, features are specifically arranged, where each feature has an associated spatial location in the image. After perceived by $\boldsymbol{U}$, features are represented by high activation values at the corresponding locations in the unit. Basically, there are two steps in our assessment of unit performance: firstly, characterize the spatial pattern hidden in these high activation values in a unit for a single image; secondly, evaluate the stability of this characterization when giving multiple image samples.

### 3.1  CHARACTERIZING THE SPATIAL PATTERN OF FEATURE REPRESENTATIONS IN A UNIT

For $\boldsymbol{U}_{i,j}$ with a grid structure, the location of an element generally refers to its coordinate index $(i, j)$. And intuitively, the spatial pattern hidden in the elements denotes certain regular relationship among their coordinate indices. So, it is natural to model such relationship with graph structure and tackle it with topological tools in the following.

**Unit and graph**  We use the edge-weighted graphs (Mehmet et al., 2019) as our basic model and construct the weighted graph $\mathcal{G} = (V, E)$ from unit $\boldsymbol{U}_{i,j}$, where $V$ is the vertex set and $E$ is the edge set. Define the adjacency matrix $\boldsymbol{A}$ of $\mathcal{G}$ as follows,

$$\boldsymbol{A} \in \mathbb{R}^{m \times m} : \boldsymbol{A}_{i,j} = \boldsymbol{U}_{i,j} \tag{2}$$

It should be noted that the individual element of $\boldsymbol{A}$ is the weight of edge in $\mathcal{G}$, which conveys the intensity of corresponding point in the $\boldsymbol{U}$.

A family of undirected graphs $\mathcal{G}^{(v)}$ with adjacency matrices $\boldsymbol{A}^{(v)}$ could be constructed by following the typical implementation of the sublevel set,

$$\boldsymbol{A}_{i,j}^{(v)} = \mathbf{1}_{\mathrm{A}_{i,j} \geq \mathrm{a}^{(v)}} \tag{3}$$

where $a^{(v)}$ is the $v$th value in the descend ordering of elements of $\boldsymbol{A}$ and $\mathbf{1}_{(\cdot)}$ is indicator function. Here, we take the adjustment of $\boldsymbol{A}^{(v)} = max(\boldsymbol{A}^{(v)}, (\boldsymbol{A}^{(v)})^T)$ to ensure the adjacency matrices $\boldsymbol{A}^{(v)}$ of undirected graphs to be symmetric.

So $\mathcal{G}^{(v)} = (V^{(v)}, E^{(v)})$ is the subgraph of $\mathcal{G}$ where $V^{(v)} = V$ and $E^{(v)} \subset E$ only includes the edges whose weights are greater than or equal to $a^{(v)}$. We have the following graph filtration,

$$\mathcal{G}_1 \subset \mathcal{G}_2 \subset \mathcal{G}_3 \subset \cdots \subset \mathcal{G} \tag{4}$$

To be more specifically, in this sublevel set filtration, it starts with the vertex set, then rank the edge weights from the maximum $a_{max}$ to minimum $a_{min}$, and let the threshold parameters decrease from $a_{max}$ to $a_{min}$. At each step, we add the corresponding edges to obtain the threshold subgraph $\mathcal{G}(v)$.

Fig.2 illustrates the construction of certain subgraph through a toy example. Consider the unit $\boldsymbol{U}_{i,j}$. We circle the locations of the top 4 largest elements in $\boldsymbol{U}_{i,j}$ (Fig.2A). Then the nonzero elements in adjacency matrix $\boldsymbol{A}^{(4)}$, $\{(1, 2), (4, 3), (2, 4), (3, 1)\}$, is located (Fig.2B) and corresponding subgraph $\mathcal{G}^{(4)}$ is constructed (Fig.2C).

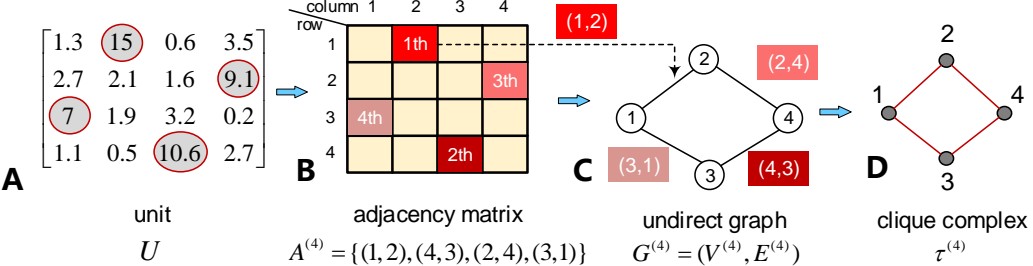

Figure 2: Example of the conversion from a unit to its clique complex.

**Complex filtration** To further reveal the correlation structure in the graphs, they are typically converted into certain kinds of topological objects, where topological invariants are calculated for capturing the high-level abstraction of correlation structure. Here, by following the common method in (Horak et al., 2009; Giovanni et al., 2013), each graph $\mathcal{G}^{(v)}$ is converted to simplicial complex (also called clique complex) $\tau^{(v)}$, as shown in Fig.2D. In this way, we have complex filtration corresponding to graph filtration (Eq.4).

$$\tau^{(1)} \subset \tau^{(2)} \subset \tau^{(3)} \subset \cdots \subset \tau \tag{5}$$

This filtration describes the evolution of correlation structure in graph $\mathcal{G}$ along with the decreasing of threshold parameter. Fig.3A shows the complex filtration of the previous example (Fig.2).

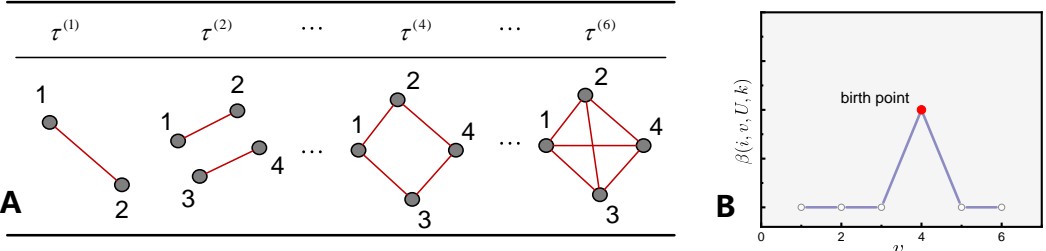

Figure 3: Instance of complex filtration (A) and Betti curve (B).

So far, we have completed the characterization from unit to the topological objects. Other than our strategy, we also discuss other alternative method, which maps the unit to the cubical complex (Kaczynski et al., 2004). See Appendix for more details.

**Betti curve and its charaterization** Next, $k$th Betti number (Hatcher, 2002) of each element in the complex filtration could be calculated using the typical computational approach of persistent homology (Ninna et al., 2017).

$$\tau^{(v)} \mapsto \beta(\tau^{(v)}) \tag{6}$$

Intuitively, $k$th Betti number $\beta(\tau^{(v)})$ could be regarded as the number of $k$-dimensional 'circle's or 'hole's or some higher order structures in complex $\tau^{(v)}$. On the other hand, many meaningful patterns in the unit would lead to the 'circle's or 'hole's of complexes in the filtration (Eq.5), see Fig.2 for illustration. In particular, the number of 'hole's is typically used as an important quantitative index for featuring such patterns. Hence, the $k$th Betti numbers $\beta(\tau^{(v)}), v \in \{1, \cdots, n\}$ could be arranged into so called $k$th Betti curves $\beta(\boldsymbol{U}, v, k)$ for the unit $\boldsymbol{U}$. Fig.3B shows the 1th Betti curve of filtration in Fig.3A.

Once having obtained the Betti curve, one needs to interpret the Betti curve and extract its core characterization. Although there exists many choices of distance between two topological diagrams such as persistence images (Adams et al., 2017), persistence landscape (Bubenik et al., 2015) and persistence entropy(Ninna et al., 2017), we find that the simple birth time of the Betti curves $\beta(\boldsymbol{U}, v, k)$ is sufficient in this characterization,

$$b(\boldsymbol{U}, k) = \inf\{v | \beta(\boldsymbol{U}, v, k) \neq 0\} \tag{7}$$

We call $b(\boldsymbol{U}, k)$ the birth time. Birth time is the indication of the critical element in complex filtration that begins to carry "hole" structure (Betti number is nonzero). It is an important sign that some essential change has occurred in complex filtration, which implies the appearance of regularized spatial pattern of notable components in the unit. Meanwhile, in some cases, no spatial pattern appear in the components in the unit, so $\beta(\boldsymbol{U}, v, k)$ constantly equals to zero, meaning that birth time doesn't exist. In general, this would happen when the unit is unable to give representations for the image, where its values are almost all zeros.

### 3.2 ASSESSING THE UNIT PERFORMANCE USING FEATURE ENTROPY

For image samples in the same class $\mathcal{C}$, an *ideal* unit $\boldsymbol{U}$ could perceive their common features. So, the spatial pattern of this unit should be similar between different image samples. In other words, the birth time obtained from each realization of units should be relatively close. That is to say, the performance of *good* unit for certain target class should be *stable* over all the samples of this class. It is the key idea for performance assessment of network unit.

**Birth distribution**  Essentially, birth time $b_{\mathcal{C}}(i, \boldsymbol{U}, k)$ is a random variable since sampling images from the specific class $\mathcal{C}$ could be regarded as statistical experiments. In fact, the probability space $(\Omega, \Sigma, P)$ could be constructed. The elements in sample space $\Omega$ are the unit $\boldsymbol{U}$ resulted from the image samples in dataset of class $C$. $\Sigma$ could be set as common discrete $\sigma$-field and probability measure $P$ is uniformly distributed on $\Omega$. In other words, every image sample has an equal chance to be chosen as the input of network model. Afterwards, $b_{\mathcal{C}}(i, \boldsymbol{U}, k)$ is defined as a random variable on $\Omega$ (where the argument is $i$, and $\boldsymbol{U}$ and $k$ are parameters),

$$b_{\mathcal{C}}(i, \boldsymbol{U}, k)(\cdot) : \Omega \to \mathbb{Z} \tag{8}$$

with the probability distribution

$$P_{\mathcal{C}, \boldsymbol{U}, k}(x) = P(b_{\mathcal{C}}(i, \boldsymbol{U}, k) = x) = \frac{b_x}{\#(\Omega)}, \tag{9}$$

where

$$b_x = \sum_{j=1}^{\#(\Omega)} \mathbf{1}_{b_{\mathcal{C}}(i, U, k) = x} \tag{10}$$

Here the composite mapping $b_{\mathcal{C}}(i, \boldsymbol{U}, k)(\cdot)$ from $\Omega$ to $\mathbb{Z}$ is composed of all the operation mentioned above, including construct weighted graphs, building complex filtration, calculating Betti curve and extracting birth time.

The degree of concentration of $P_{\mathcal{C}, \boldsymbol{U}, k}(x)$ gives a direct view about the performance of unit $\boldsymbol{U}$ on class $\mathcal{C}$, as illustrated in Fig.1. More specifically, if the distribution presents close to a degenerate-like style, it means that the underlying common features of the class $\mathcal{C}$ could be *stably* perceived by the unit $\boldsymbol{U}$. On the contrary, the distribution presents close to a uniform-like style when features are perceived almost blindly, indicating that unit $\boldsymbol{U}$ is invalid for $\mathcal{C}$. In summary, the degree of concentration of $P_{C, \boldsymbol{U}, k}(x)$ is supposed to be an effective indicator of the performance of unit $\boldsymbol{U}$.

**Feature entropy**  To further quantize the degree of concentration of birth distribution $P_{\mathcal{C}, \boldsymbol{U}, k}(x)$, we introduce its entropy $H_{\mathcal{C}, \boldsymbol{U}, k}$ and call it feature entropy,

$$H_{\mathcal{C}, \boldsymbol{U}, k} = -\sum_x P_{\mathcal{C}, \boldsymbol{U}, k}(x) \log P_{\mathcal{C}, \boldsymbol{U}, k}(x) \tag{11}$$

It should be noted that the birth time in Eq.7 may not exist for some input images in class $\mathcal{C}$ and unit $\boldsymbol{U}$. For unit $\boldsymbol{U}$, the percentage of images in class $\mathcal{C}$ having birth times, termed as selective rate $\epsilon_{\mathcal{C}, \boldsymbol{U}}$, is also a crucial factor to the effectiveness of $\boldsymbol{U}$ on $\mathcal{C}$. If the $\epsilon_{\mathcal{C}, \boldsymbol{U}}$ is too low, it indicates that the unit could not perceive most of the image samples in this class. In this situation, extremely low $\epsilon_{\mathcal{C}, \boldsymbol{U}}$ would cause the feature entropy approach to zero, but the unit should be judged as completely invalid. Therefore, we rule out this extreme case by setting a threshold $p$ and for completeness, and assign the feature entropy associated with the maximum of feature entropy $\Omega$ for the set of samples,

$$H_{\mathcal{C}, U, k} = \begin{cases} H_{\mathcal{C}, U, k} & \epsilon_{\mathcal{C}, \boldsymbol{U}} \geq p \\ (1 - \epsilon_{\mathcal{C}, \boldsymbol{U}}) \cdot \log |\Omega| & \epsilon_{\mathcal{C}, \boldsymbol{U}} < p \end{cases} \tag{12}$$

Here, $p$ is prescribed as 0.1 in our computation.

## 4 EXPERIMENTS

For experiments, we use the VGG16 network architecture to perform the image classification task on the ImageNet dataset. Unless otherwise stated, the exampled VGG16 model is trained from scratch with the hyperparameters deployed in Simonyan & Zisserman (2014). For clarity, we only calculate birth times $b_{\mathcal{C}}(i, U, 1)$ based on 1th betti curve for all the units. Also, it should be noted that our method focuses on the behaviors of feature extraction operations and has not utilized any kind of particular nature of VGG network architecture, and all our investigation could be applicable to other network architectures effortlessly.

### 4.1 CALCULATION FLOW

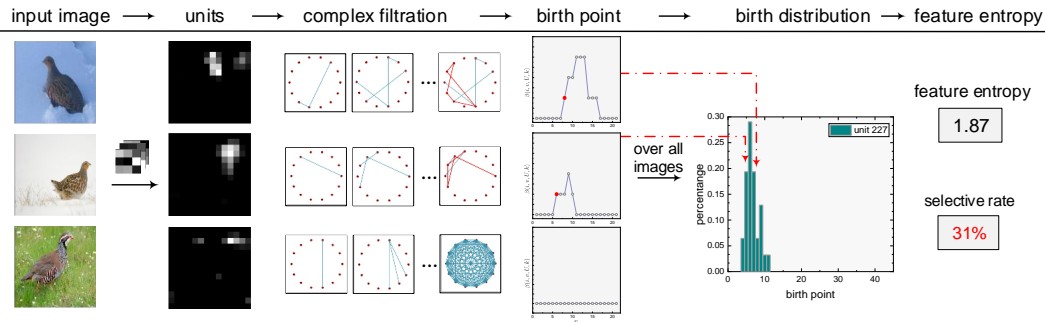

Figure 4: Calculation flow of feature entropy.

As an example, the class partridge (wnid n01807496) in ImageNet is chosen for illustration. Here, we sample 100 images from its training set as the image set for building the birth distribution. Fig.4 shows the calculation flow. It starts from extracting all the units for each image sample. By characterizing the unit with graph model, each unit corresponds to a specific filtration. Then, using formular 7, we can obtain the birth time of each unit. In this way, the distribution of birth time could be set up via Eq.9 over the sampled images. Fig.4 shows the histogram of the birth time distribution for a specific unit in the the last convolution layer "block5_conv3". Likewise, the feature entropy can be calculated via Eq.11 for all other units.

### 4.2 LAYER AND TRAINING ANALYSIS

**Layer analysis** Here, we check the status of units in each convolutional layer, where we average the feature entropy across all the units within the layer to indicate the overall status of units in this layer. Using the same image set in the previous section, Fig.5A(1-2) give comparisons of results between the convergence model and the random-initialized model.

In Fig.5A(1), we could clearly see that for the convergence model, the feature entropy continually decrease as the layers go deeper. This is as expected because as the layer goes deeper, units are considered to perceive more advanced features than previous layers, so the spatial pattern in these features would be more significant. As for the random-initialized model, since units are incapable to perceive the common features, we could not observe a clear decrease of feature entropy, and meanwhile the feature entropy in every layer is higher than that in the convergence model. In Fig.5A(2), we could also find that each layer in the convergence model would present a higher selective rate than the corresponding layer in the random-initialized model, except for the last convolutional layer "block5_conv3". Also, the selective rate of the last convolutional layer is much more lower than other layers. The low feature entropy and fairly high selective rate indicate that comparing to units in other layers, units at the last convolutional layer in the convergence model would present strong specialization and exhibit the most effective representations of features to this class.

Then, we randomly choose 100 classes in ImageNet and average the feature entropy across all these classes on the convergence model. Fig.5A(3) shows the results. We could see that the results are very similar to Fig.5A(1-2), which confirms the fact further.

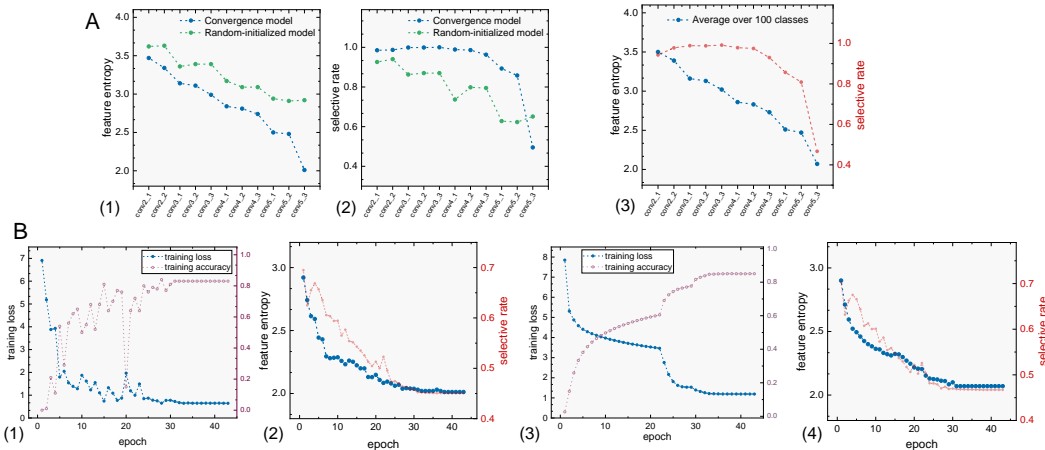

Figure 5: (A) Comparisons of feature entropy (1) and selective rate (2) of different layers between the convergence model and random-initialized model, where (3) shows the results over 100 classes. (B) Simultaneous evolution of training loss and feature entropy during training for the chosen class (1-2) and for the 100 classes (3-4).

**Training analysis** Then, we investigate the variation of feature entropy of the last convolutional layer during training. Fig.5B(1-2) show the results on the same example class used previously, and Fig.5B(3-4) show the results across 100 classes chosen previously. In both situations, we could find that the feature entropy would decrease during training, indicating that units are gradually learned to be able to perceive the common features in the class. And remarkably, the decreasing pattern of feature entropy and that of training cross-entropy loss coincide approximately. Both of them experience a comparable big drop in the first epoch and gradually down to the convergence level. This means that feature entropy is a valid indicator of network performance.

### 4.3 INDICATOR OF STATUS OF NETWORK UNIT

To investigate the ability of feature entropy as indicator of unit status, we make comparisons with some commonly-used analogous network indicators including L1-norm (Li et al., 2016), APoZ (He et al., 2017b), and a more generalized form of class selectivity used in Zhou et al. (2018b). Here, the unit and the image set in the previous subsection are still used in the following demonstration.

**Rescaling investigation** The comparison is implemented by rescaling the magnitude of values to half for all the input images or all the CNN filters connecting to the layer. Both the two implementations could potentially cause the values in units within the layer vary with the same scale, but in general have no substantial impact on the network performance and the function of each unit. In other words, units should be indicated as almost the same with or without such implementation.

Table 1: Comparisons of unit status by rescaling the values in images or CNN filters

| Images | CNN filters | Accuracy | L1-norm | APoZ | Class selectivity | **Feature entropy** |
|---|---|---|---|---|---|---|
| ✕ | ✕ | 0.83 | 29.5 | 17.14% | 0.58 | 1.87 |
| Half scale | ✕ | 0.81 | 14.7 | 17.15% | 0.31 | 1.90 |
| ✕ | Half scale | 0.83 | 14.6 | 17.14% | 0.30 | 1.87 |
| Half scale | Half scale | 0.79 | 7.2 | 17.16% | 0.03 | 1.92 |

Table 1 shows the results where ✕ denotes no rescaling operation for the item. As half scaling the magnitude in input images or units, the performance of the model fluctuates slightly. We could find that APoZ and feature entropy vary in the similar way with the performance, but L1-norm and class selectivity vary terribly. Apparently, despite little effect for the network, rescaling operations would have a major impact on these magnitude-based indicators, like L1-norm and class selectivity. These indicators fail to give accurate and stable measure of unit status especially when facing images or units with different value scales.

Table 2: Comparisons of unit status with respect to well-trained units and random units

|  | L1-norm | APoZ | Class selectivity | **Feature entropy** |
|---|---|---|---|---|
| Well-trained unit | 29.5 | 17.14% | 0.58 | 1.87 |
| Random initialized units | 32.2(30.9) | 41%(40%) | 0.01(0.003) | 2.87(0.21), 0.83(0.22) |

**Detecting randomness in units**  Next, we compare the status of this unit with random units (units yielded by random-initialized models). Table 2 presents the results. The random units are sampled 100 times and the presented results are averaged over the 100 samples where the value in the brackets denotes the standard deviation. Since random units are clearly incapable to perceive features well like those trained units, they are expected to be indicated as ineffective units. We could see that when using L1-norm and APoZ indicators, they are impossible to give a stable indication as the standard deviation is extremely large. In some samples, the random units are judged as much "better" than the trained units, which is obviously incorrect. Accordingly, it could be also misleading using APoZ as the indicator of unit status. In contrast, the feature entropy would consistently be very high when random pattern exists in the unit, providing a well discrimination between trained units and random ones.

## 4.4 USING FEATURE ENTROPY TO INDICATE NETWORKS WITH DIFFERENT GENERALIZATION

In general, due to the large hypothesis space, CNNs could converge to a variety of minima on the dataset. Since feature entropy could reliably indicate the status of network units, it is natural to use it to discriminate which minima could provide more effective feature representations.

**Models**  In this subsection, we prepare two sets of VGG16 models. Model set A consists of four models trained from scratch with different hyperparameters on ImageNet dataset, and Model set B consists of five models trained from scratch to almost zero training error with the same hyperparameters but on ImageNet dataset with different fractions of randomly corrupted labels as introduced in Zhang et al. (2017). Table 3 and 4 separately show the performance of models in the two model sets. In model set B, we use Model AD in Model set A as the first model Model BA with no corruption rate. Besides, it should be noted that all the calculation in this section is based on the image sampled from the training dataset.

Table 3: Model set A

|  | Train Acc | Test Acc |
|---|---|---|
| Model AA | 0.732 | 0.657 |
| Model AB | 0.818 | 0.532 |
| Model AC | 0.828 | 0.444 |
| Model AD | 0.996 | 0.378 |

Table 4: Model set B

|  | Train Acc | Test Acc | Corrupted |
|---|---|---|---|
| Model BA | 0.996 | 0.378 | 0.0 |
| Model BB | 0.992 | 0.297 | 0.2 |
| Model BC | 0.994 | 0.166 | 0.4 |
| Model BD | 0.992 | 0.074 | 0.6 |
| Model BE | 0.993 | 0.010 | 0.8 |

**Model set A**  Using the same image set in previous section, we start by investigating the feature entropy of units at different layers in the four models. Here, we still use the averaged feature entropy across all the units within a layer to indicate the overall level of how well the units in this layer could perceive the features. Fig.6A(1-2) shows the results of this class. We could see in the figure that there would not be significant difference of feature entropy between these models in layers except for the last convolutional layer. And in the last convolutional layer, for models with better generalization, their feature entropy would be lower than those with poor generalization, indicating that they would provide more effective feature representations. Besides, as for the selective rate, the four models are quite close.

Then, we randomly choose 100 classes in the ImageNet dataset and calculate the feature entropy of the units in the last convolutional layer. Fig.6A(3) presents the scatter plot for the four models, where each point stands for the feature entropy and selective rate of a specific class. For each model, its points locate at an area separately from other models, giving a discrimination between models. Also similarly, models with better generalization have points with lower feature entropy.

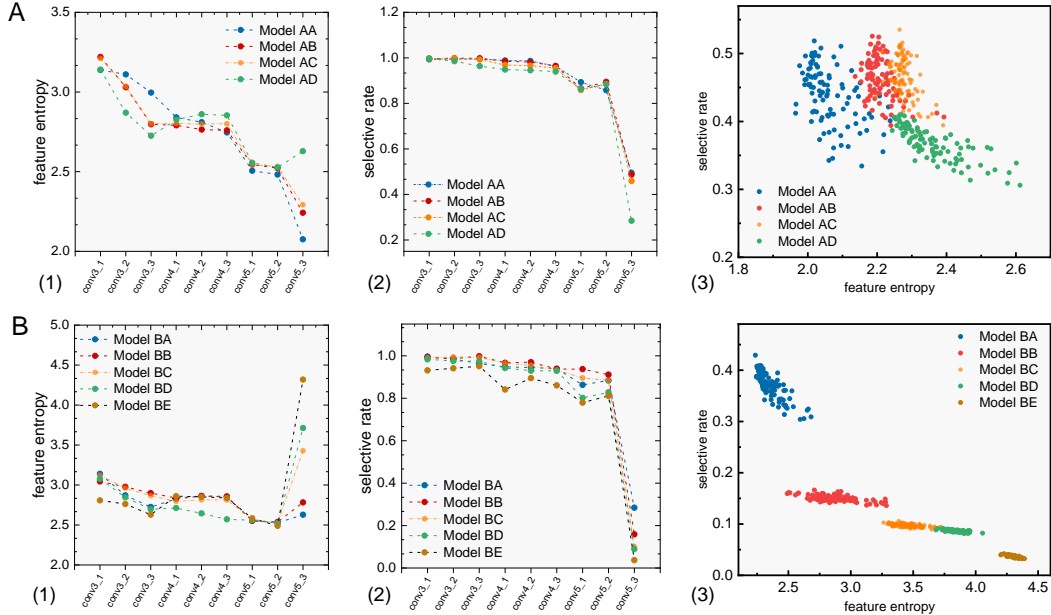

Figure 6: Comparisons between models in separately model set A (A) and model set B (B). Compare the feature entropy (1) and selective rate (2) of units at different layers between models in the corresponding model set on the exampled class. (3) shows the scatter plot between feature entropy and selective rate of units at the last convolutional layer on the 100 sampled classes.

**Model set B** For model set B, we use the same implementation as applied previously in the model set A, where the results are shown in Fig.6B. Comparing to the Model set A, since using the partially corrupted labels, units in the Model Set B are unable to perceive the common features between samples in the same class, which causes that the selective rate of most units are extremely low as shown in Fig.6B(2). Due to such low selective rate, we could also find in Fig.6B(1) that feature entropy of the units in the last convolutional layer may abruptly reach to a very high point. The more fraction the labels are corrupted, the higher feature entropy the units are and in the meantime the lower the selective rate the units are. This could be observed as well in Fig.6B(3) where the 100 classes are used for calculation.

## 5 CONCLUSION

We propose a novel method that could give quantitative identification of individual unit status, called feature entropy, for a specific class using algebraic topological tools. We show that feature entropy is a reliable indicator of unit status that could well cope with various cases such as rescaling values or existence of randomness. Also we show that feature entropy behaves in the similar way as loss during the training stage and presents a descending trend as convolutional layers go deeper. Using feature entropy, we show that CNNs with different generalization could be discriminated by the effectiveness of feature representations of the units in the last convolutional layer. We suppose this would be helpful for further understanding the mechanism of convolutional neural networks.

### ACKNOWLEDGMENTS

We would like to thank Brain-Inspired Research Team at Tsinghua University for the discussions. We would like to thank the reviewers for their helpful comments.

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

# A  APPENDIX

## A.1  IMPLEMENTATION DETAILS OF MODELS

In our experiments, the networks we used are the standard VGG16 architecture. We simply resize all the mentioned sample images to $224 \times 224$ without implementing any data augment (like random crop) during all the experiments (including the feature entropy calculation and the reported performance). The model used for demonstration in the experiment section (besides the last subsection) is Model AA in model set A.

For model set A, we use the following implementations,

- Model AA. The hyper-parameters are the same with them in the paper (Simonyan & Zisserman, 2014).
- Model AB. The hyper-parameters are the same with them in Model AA, except for changing the momentum to 0 and without using the data augmentation strategy.
- Model AC. The hyper-parameters are the same with them in Model AB, except for that only the first fully connected layer use the dropout with the rate of 0.3.
- Model AD. None of the conventional training enhancement technique is applied. Basically, It is Model AC without using dropout and l2 regularization.

For model set B, we use the following implementations,

- Model BA. It is actually the Model AD.
- Model BB. The hyper-parameters are the same with them in Model BA, except for corrupting the labels with 0.2 fraction.
- Model BC. The hyper-parameters are the same with them in Model BA, except for corrupting the labels with 0.4 fraction.
- Model BD. The hyper-parameters are the same with them in Model BA, except for corrupting the labels with 0.6 fraction.
- Model BE. The hyper-parameters are the same with them in Model BA, except for corrupting the labels with 0.8 fraction.

## A.2 Spatial Characterization using Cubical Complex

In our method, a unit is firstly convert to a set of graphs and then each graph could be converted to the corresponding clique complex. In this way, a unit is characterized by the filtration of clique complexes. Alternatively, the unit could be directly modeled as a cubical complex (Kaczynski et al., 2004), and then use the similar sublevel set implementation to acquire the filtration of cubical complexes. So here, we use cubical complex instead of clique complex and meanwhile keep the rest of the methods unchanged.

By investigating the birth distribution on the classes in the Imagenet with Model AA, we found that for every unit especially at relatively deeper layers, almost 95% images would have no birth time. In other words, no persistence homology emerges for almost all the units when using cubical complex. Thus, it is impossible to give a further calculation of the stability for images in the same class.

## A.3 Discussions of using Birth Time

When characterizing the filtration of topological complexes, we use the birth time of the filtration. Comparing to some other characterizations, we suppose that the advantages of using birth time may lie in,

- Birth time is very easy to compute. In practice, when using birth time, it is not necessary to go through the whole filtration process, where we could stop the calculation when the birth time emerges. In this way, we could largely save the computation especially when the size of units is very high. This could be very helpful for calculating the large amount of units in neural networks.

- Birth time is very convenient for the investigation of stability between samples because it is an integer essentially. So, the discrete distribution could be set up effortlessly. Besides, when using entropy to investigate the stability of birth times, the results are bounded strictly in the range from 0 to $\log N$.

- Birth time is suitable for the idea. Since the significant features are generally represented by the high activation values in the units, it is more expected that the regularized pattern could be formed by these values. So it is natural to focus on these high activation values, which leads to the use of birth time.

Then, considering the various topological features, we compare the effect of using birth time and two other characterizations of the Betti curves which are its maximum value and its integration. During calculation, we would only change birth time to the use of given characterization, where all the other parts remain the same. Table 5 shows the results. We find that the difference between using birth time and using other characterization are acceptable. But when using other characterizations, we need to calculate the whole Betti curve. The computation cost may be hundreds times than using birth time.

Table 5: Comparisons of feature entropy by using different characterizations

|  | Birth time | Maximum | Integration |
|---|---|---|---|
| Reference unit | 1.87 | 1.89 | 1.92 |
| Last convolutional layer | 2.01 | 2.09 | 2.11 |
| Last convolutional layer (100 classes) | 2.07 | 2.13 | 2.17 |

## A.4 Study on Sample Size

For an efficient computation in practice, we could use part of the training set for calculating the feature entropy. In this section, we check the influence on the feature entropy when using different sample sizes.

We test the sample size from the 50 to the full size of of the example class (about 1300 images) in the training set on the reference model (Model AA) used in the paper. We first investigate the variation of feature entropy of a single unit when changing the sample size. We performed 100 times of

sampling to investigate the stability of feature entropy, and each time randomly sample 500, 100 and 50 images from the training set. Fig.7 shows the histogram of feature entropy of the unit used in Section 4.1 for the 100 times of sampling, where the x-axis is feature entropy and y-axis is the frequency. We could see that for 500 and 100 samples, their feature entropy distribute closely to the value of feature entropy for using the whole training set. However, as for 50 samples, the feature entropy distribute scatteredly. Therefore, using 100 samples could give considerable feature entropy of the class and in the meantime largely reduce the computation cost.

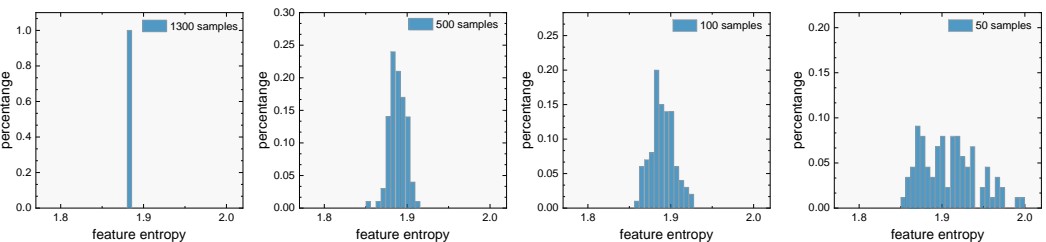

Figure 7: Histogram of the feature entropy in 100 trials for different sample size.

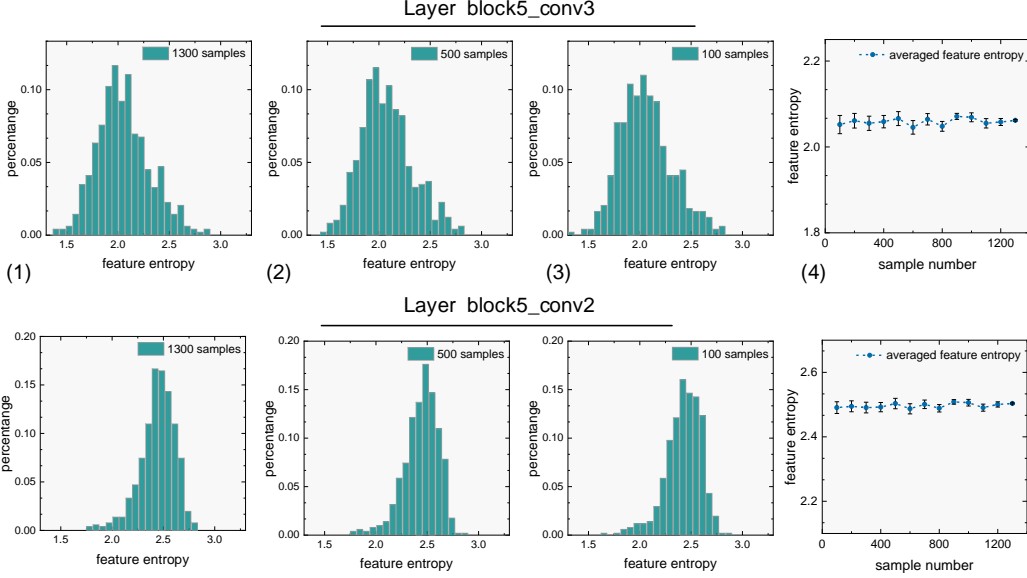

Figure 8: (1-3) Histogram of the feature entropy of units in the layer "block5_conv3" and "block5_conv2" for different sample size.(4) Feature entropy averaged across all the units in the layer with respect to the sample size. Error bars stand for performing the sampling 100 times.

Next, we investigate the feature entropy of all the units at a layer when changing the sample size. All the units at the layer "block5_conv3" and "block5_conv2" in the reference model are used, and Fig.2A presents the results. As we could see in Fig.8(1-3), for 1300, 500 and 100 samples, their distributions of feature entropy of the 512 units at the layer are very close. And in Fig.8(4), we average the feature entropy across the 512 units, and we could see that the feature entropy vary slightly from using 100 samples to 1300 samples. Besides, the error bar is acquired via performing the 100 times of sampling. And we could see the error bars decrease as the sample sizes increase, and even for 100 samples, its error bar is still very low.

## A.5 ADDITIONAL RESULTS ON RESNET

In this section, we perform the experiments on the ResNet34, which is trained from scratch with the hyperparameters deployed in (He et al., 2016).

**Rescaling and randomness comparison**  For rescaling and randomness investigation, the experiment setups are the same as those in Section 4.3. Here, the unit is chosen at the layer "conv5_3_out", which is the last layer before global pooling layer. The suffix "out" stands for the output of a residual convolutional block. Table 6 shows the corresponding results when performing rescaling operation.

Table 6: Comparisons of unit status by rescaling the values in images or CNN filters

| Images | CNN filters | Accuracy | L1-norm | APoZ | Class selectivity | **Feature entropy** |
|--------|-------------|----------|---------|------|-------------------|---------------------|
| ✗ | ✗ | 0.81 | 122.1 | 4.79% | 0.47 | 1.71 |
| Half scale | ✗ | 0.80 | 60.9 | 4.81% | 0.23 | 1.72 |
| ✗ | Half scale | 0.81 | 61.1 | 4.79% | 0.24 | 1.71 |
| Half scale | Half scale | 0.77 | 30.4 | 4.83% | 0.07 | 1.74 |

Table 7 shows the corresponding results of trained units and random initialized units.

Table 7: Comparisons of unit status with respect to well-trained units and random units

| | L1-norm | APoZ | Class selectivity | **Feature entropy** |
|--------|---------|------|-------------------|---------------------|
| Well-trained unit | 122.1 | 4.79% | 0.47 | 1.71 |
| Random initialized units | 678(317) | 1.17%(1.54%) | 0.02(0.013) | 2.08(0.29) |

Just as the results on VGG16 (shown in Table 1 and 2), due to the advantages of using topology, feature entropy could give stable indication of the status of units as expected, no matter with the scaling operation or in the randomness situation.

**Layer and training analysis**  We follow the implementation in the VGG16 network (Section 4.2) and first check the status of unit in each convolution layer in ResNet34. Fig.9 gives the comparisons between the results of the convergence ResNet34 model and the random-initialized model. For Fig.9A, it shows the variation of feature entropy with respect to the output of each convolutional block in the ResNet34 model. We could see that the feature entropy continually decrease as the layer goes deeper, similar to the results on VGG16. Besides, the feature entropy of each layer in the random-initialized model are all larger than the corresponding layer in the convergence model.

Then, we check the variation of feature entropy for layers in a convolutional block, where Fig.9B shows the results. Typically, for ResNet34 architecture, a convolutional block consists of two consecutive convolutional layers and yields the output after the shortcut connection. We could see in the figure that the feature entropy at the first convolutional layer would increase at the second convolutional layer, which is because of the non-activation of units at the second convolutional layer. After that, the shortcut connection would decrease the feature entropy even without being activated and finally reach at a lower value than that at the first convolutional layer. The shortcut connection plays an important role in making the features more significant, which leads to the decrease in the feature entropy.

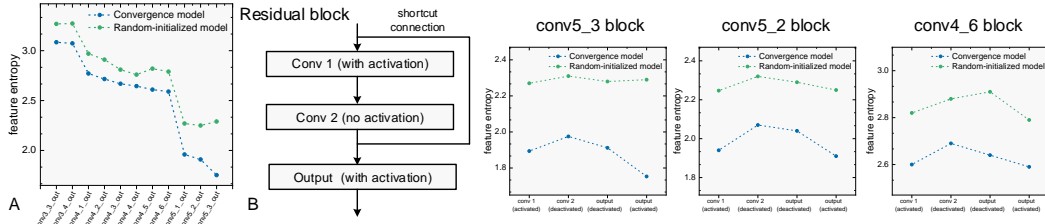

Figure 9: (A) Comparisons of feature entropy of different layers between the convergence model and random-initialized model. (B) Comparisons of feature entropy of the layers in separately three residual blocks between the convergence model and random-initialized model.

Next, Fig.10 shows the variation of feature entropy during training. Similarly to those on the VGG16 model, the feature entropy decrease during training and behaves close to the training loss as well.

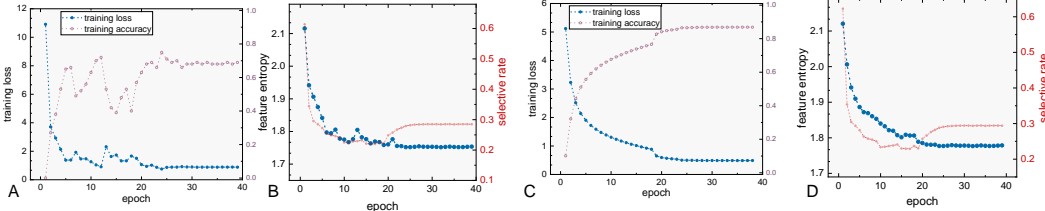

Figure 10: (B) Simultaneous evolution of training loss and feature entropy during training for the chosen class (A-B) and for the 100 classes (C-D).

**Indicating models with differnt generalization** Further, a set of ResNet34 models with different generalization is trained via the same partially corrupted strategy as in Section 4.4. Table A.5 shows the corresponding performance and Fig.11 gives the results of feature entropy. As we could see in the figure, results on ResNet model are quite similar to those on VGG16 (Fig.4.4). The feature entropy of models would gradually increase as the generalization become worse.

Table 8: Performance of model set R

|  | Train Acc | Test Acc | Corrupted |
|---|---|---|---|
| Model RA | 0.823 | 0.714 | 0.0 |
| Model RB | 0.982 | 0.382 | 0.2 |
| Model RC | 0.991 | 0.229 | 0.4 |
| Model RD | 0.995 | 0.102 | 0.6 |
| Model RE | 0.991 | 0.036 | 0.8 |

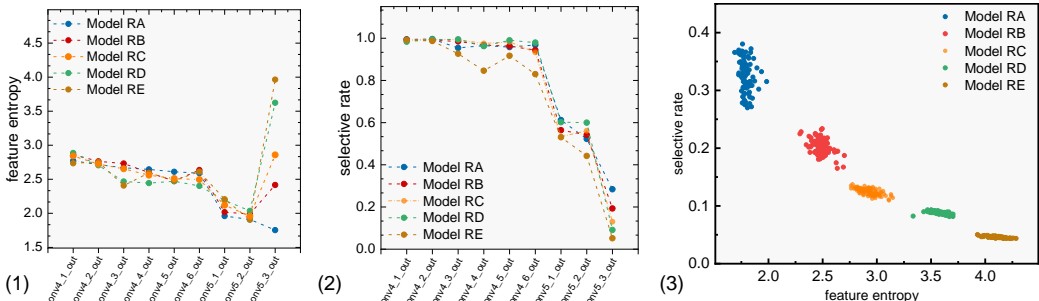

Figure 11: Comparisons between models in model set R. Compare the feature entropy (1) and selective rate (2) of units at different layers between models in the corresponding model set on the exampled class. (3) shows the scatter plot between feature entropy and selective rate of units at the last residual block on the 100 sampled classes.

## A.6 COMPARISONS WITH OTHER RELATED METHODS USED IN PRUNING

In this section, we compare feature entropy with two other related strategies used in pruning. One is filter pruning via geometric median (FPGM) (He et al., 2019), which uses the norm distance to the geometric median of the set of filters as the indicator of the importance of a filter $F_i$ at a layer (Eq.4 in He et al. (2019)),

$$g(F_i) = \sum_{F_j \in \{F_k\}_{k=0}^{N}} \|F_i - F_j\|_2 \tag{13}$$

where $N$ denotes the total number of filters at the layer. A filter is considered as useless if its $g(F_i)$ is small.

The other one is called neuron importance score propagation (NISP) (Yu et al., 2018), which measures the importance of a unit by propagating the final feature selection score backwards based on coefficients of network weights at a specific layer (Eq.19 in Yu et al. (2018)),

$$s_{k,i} = \sum_j |\boldsymbol{W}_{i,j}^{(k+l)}| s_{k+1,j} \tag{14}$$

where $|\cdot|$ is the element-wise absolute value, $s_{k,i}$ denotes the importance score for unit $i$ at layer $k$ and $\boldsymbol{W}_{i,j}^{(k+1)}$ denotes the weight matrix at layer $(k+1)$. The method needs a base metric to score the final feature selection and use it to calculate units in other layers backwards via this formula. Here, we follow the metric used in the original paper called infinite feature selection (Roffo et al., 2015).

We follow the same investigations of rescaling operations and randomness detection as deployed previously in Section 4.3. Table 9 and Table 10 show the corresponding results. We could see in the Table 9 that just as the other methods in Section 4.3, rescaling operations would have a major impact on the two indicators. Apparently, from the above formulas of the two methods, we could easily find that they are still essentially based on the magnitude of weights. Therefore, it is nature that they fail to give valid indication in these complicated situations.

Table 9: Comparisons of unit status by rescaling the values in images or CNN filters

| Images | CNN filters | Accuracy | FPGM | NISP | Feature entropy |
|--------|-------------|----------|------|------|-----------------|
| ✕ | ✕ | 0.83 | 0.88 | 72.37 | 1.87 |
| Half scale | ✕ | 0.81 | 0.88 | 36.04 | 1.90 |
| ✕ | Half scale | 0.83 | 0.44 | 36.17 | 1.87 |
| Half scale | Half scale | 0.79 | 0.44 | 17.81 | 1.92 |

In addition to the rescaling operation, as we could see in Table 10, the FPGM method is also unable to give correct discrimination between the well-trained units and the random-initialized units.

Table 10: Comparisons of unit status with respect to well-trained units and random units

| | FPGM | NISP | Feature entropy |
|---|------|------|-----------------|
| Well-trained unit | 0.88 | 72.37 | 1.87 |
| Random initialized units | 1.44(0.044) | 19.27(2.98) | 2.87(0.22) |

## A.7 STUDY OF PRUNING

In Section 4.4, feature entropy has shown the ability to reliably indicate the status of units between different models in various situations, which is the core motivation of the paper. In addition, we also show the effectiveness of feature entropy to give comparisons between different units in the single model in a fixed situation. In this situation, since units are generally in the comparable scale (unless use some specific implementations on networks like Neyshabur et al. (2015)[2]), the mentioned problems may be largely alleviated, so units would be compared directly via the typical methods like L1-Norm, etc. Besides, we would consider feature entropy and selective rate both to represent its effectiveness of units for an accurate estimation, where we would fuse these two factors by using $H/\epsilon$ in the following calculations.

Here we are going to investigate the units in the single model in two parts. The first part is the cumulative unit ablation and the second part is an example of the implementation of network pruning.

### A.7.1 CUMULATIVE UNIT ABLATION

**Ablation test setting** For a given class, cumulative unit ablation tests check the evolution of the network performance by progressively removing each unit within a layer according to the order of

---

[2]This implementation could somehow yield the same effect with performing the rescaling operation to the magnitude of a specific unit in Section 4.3, without changing the results of networks and other units. For more details about this implementation, please refer to the paper.

certain kind of sorted attribute of units. Typically, removing a unit refers to forcing the outputs of the unit to all zeros. Here, the unit's feature entropy is chosen as the attribute, and descending and ascending orders are considered simultaneously. According to our idea, the unit with lower feature entropy is more effective than that with higher feature entropy. The cumulative ablation tests reveal the relation between feature entropy of unit's output and its effectiveness on network performance, where training sets are generally used for calculating the feature entropy of the units and testing sets are used for checking their impact on the network performance. Besides, the investigations are still using the last convolutional layer on the same image set in the previous section. In particular, for each image in the set, it would be randomly rescaled within the ratio from 0.5 to 1.5.

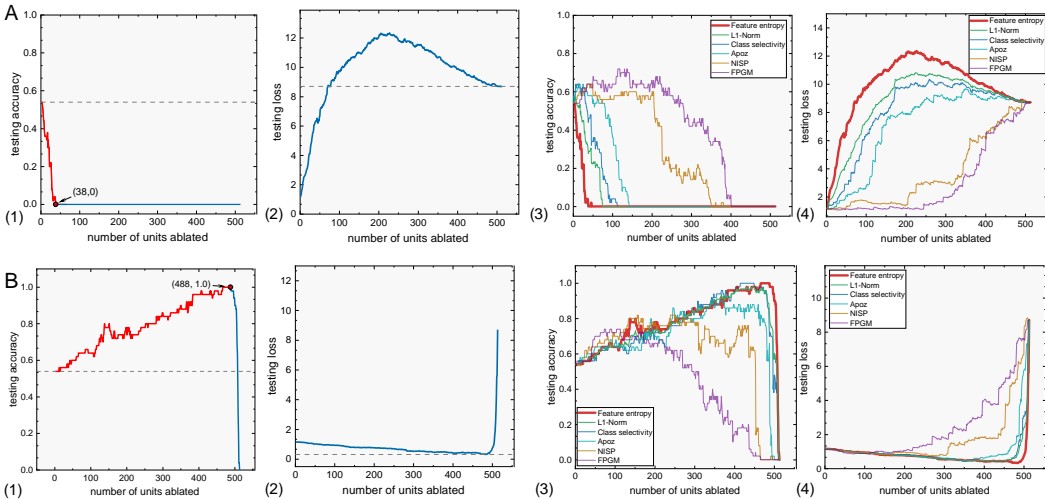

Figure 12: Cumulative ablation curves of accuracy (1) and loss (2) according to the descending rank (A) and the ascending rank (B), together with comparisons with other methods (3-4).

**Results** The cumulative ablation is performed via ascending order of feature entropy first. Fig.12A shows the variation of the testing accuracy (1) and loss (2) during ablation. We can see the accuracy drops rapidly in the beginning and arrives at zero after ablating about 38 units. On the other side, the loss sharply increases in the beginning just like accuracy, afterwards slowly climbing and reaching its peak value (12.32) during ablation. It should be mentioned that the peak value even exceeds that of removing all the units (8.71). Being somehow counterintuitive, the ablation has generated a layer even worse than the all-zero layer.

We have also compared with the performance evolutions resulted from APoZ, class selectivity, L1-norm, FPGM, NISP. We can see in Fig.12(3-4) that the accuracy drops (or the loss increases) more rapidly using feature entropy (red line) than others during the ablation process. This implies that those most effective units picked by feature entropy would be more impactful to the network performance.

Then, the cumulative ablation is performed via descending order of feature entropy. From Fig.12B, instead of decreasing, the testing accuracy features a slow increase from 0.54 to surprising 1.0 when 96% of the units are removed. These units do not lie in the head part of feature entropy rank and are considered as the less effective units. So removing them will promote the accuracy and enhance the network performance. Remarkably, starting with removing a certain unit, the accuracy dramatically drops to zero within only 30 units. It means that these units are critical and removing them will lead to breakdown of network. Likewise, the comparisons are also implemented with other attributes in Fig.12A. We could also find that curve of feature entropy could reach much higher peak point in accuracy, which shows the advantage of feature entropy.

Interestingly, we could see that the testing performance could be largely enhanced via the cumulative unit ablation for a single class. Then, all the 1000 classes in ImageNet are investigated in the same way. On the one hand, we would check whether the testing performance could be enhanced

Table 11: Performance enhancement via cumulative ablation when using different methods

|  | Unchanged network | Ablated network | Performance enhanced |
|---|---|---|---|
| L1-Norm | 0.759[1] | 0.928 | 0.167 |
| Apoz | 0.759 | 0.873 | 0.114 |
| Class selectivity | 0.759 | 0.917 | 0.158 |
| FPGM | 0.759 | 0.789 | 0.030 |
| NISP | 0.759 | 0.838 | 0.079 |
| **Feature entropy** | 0.759 | **0.945** | **0.186** |

[1] All the performances are reported as balanced accuracy (Brodersen et al., 2010).

in this way for the majority of the classes, on the other hand, we use this to give comparisons between different methods under this circumstance. Table 11 shows the corresponding results, which is reported by averaging across all the classes. Note that all the results in the table are reported as the balanced accuracy, which is commonly used for assessing the performance where the task is imbalanced. We could see that after performing ablation on a single class, the network could generally acquire performance enhancement to some extent. And compared to other methods, feature entropy shows a larger performance enhancement, which again demonstrates the superiority of our method.

### A.7.2 IMPLEMENTATION OF NETWORK PRUNING

In the previous paragraph, we perform the ablation test on a single class to show the effectiveness of feature entropy. Here, we are going to give implementation of channel-level network pruning for the whole dataset via feature entropy.

**Network pruning setting** The network is pruned by following the layer-by-layer strategy, which prunes the channels from the shallow to the deep layers progressively in two stages. In the first stage, unimportant units at a layer are selected to be pruned based on a given pruning ratio via feature entropy. For units at the layer, they are selected by averaging the feature entropy across numbers of classes,

$$H(\boldsymbol{U}_i) = \frac{1}{K} \sum_{k=1}^{K} H_k(\boldsymbol{U}_i) \tag{15}$$

where $K$ is the total number of chosen classes. In this way, we would prune the corresponding filters whose outputted units have higher feature entropy. In the meantime, since the unit is removed, channels of the filters that connect this unit and units at next layer would also be pruned. Then in the second stage, we would fine-tune the pruned model. Here, we still use the reference VGG16 model in the paper as our target network. In the conventional VGG architecture, after the convolutional backbone, two fully connected layers are used as the classifier for the following decision making. Although parameters in the fully connected layers could account for over 80%, yet the majority of computation cost lie on the convolutional operation. So here, we remain this classifier and target to prune all the convolutional layers for computation reduction.

Additionally, during fine-tuning stage, we use the SGD optimizer with a learning rate which is initially set to 1e-3, decay 10 times after one epoch fine-tuning and finally stop after 1e-5. All implementations are deployed on the Nvidia-A100 station with a batch size of 512. Also, similar to the setting in the paper, all the images are simply resized to $224 \times 224$. We use the FLOPs of convolutional operation as the metric of computation cost and parameter counts as the metric of storage cost.

**Results** We first investigate the impact of the number of classes $K$ to the pruning effect, where the pruning ratio is set to 50%. Fig.13A shows the pruning results of VGG16 with remaining fully connected layers. We could see that as we choose more amount of classes for the calculation of

feature entropy, the accuracy of the fine-tuned model gradually increase. And compared to using 100 classes, using 150 and 200 classes would increase the accuracy slightly. Thus, for an efficient computation of feature entropy, we would use 100 classes for the following pruning implementation.

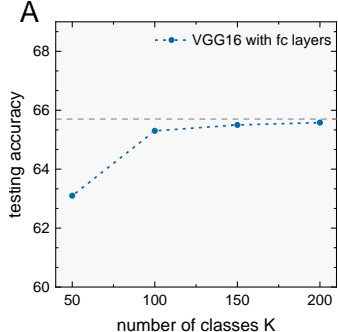 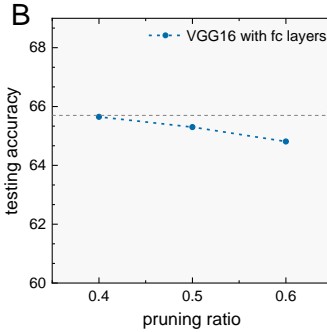

Figure 13: (A) Accuracy of the fine-tuned models with respect to the number chosen classes used for feature entropy calculation. (B) Accuracy of the fine-tuned models on different pruning ratios.

Then, we give the results of fine-tuned networks via the implementations of separately 40%, 50%, 60% pruning ratios, as shown in Fig.13B. We could see that as we keep more channels, the fine-tuned network could reach a higher accuracy and when the pruning ratio is 60%, the accuracy is comparable to that of unchanged network.

Finally, we make comparisons when using different methods with the pruning ratio of 50%. For other methods, we use the same pruning implementation as feature entropy. Table 12 gives the final results. As we could see in the table, the model pruned via feature entropy could reach competitive results compared to other methods. In conclusion, feature entropy shows the advantage of being an effective indicator for not only network units between different models but units in a single model.

Table 12: Comparisons of the accuracy of fine-tuned models when using different methods

| | Accuracy ↓ (fine-tuned) | Pruned ratio | #Params[1] | FLOPs[1] | img/sec[2] |
|---|---|---|---|---|---|
| Unchanged network | 65.71% | - | 138M | 30.96B | 2,178 |
| L1-Norm | 1.03% | | | | |
| Apoz | 1.59% | | | | |
| Class selectivity | **0.38%** | 50% | 75.9M | 7.87B | 5,541[3] |
| FPGM | 0.59% | | | | |
| NISP | 0.92% | | | | |
| | **0.09%** | 40% | 87.8M | 11.15B | 4,869 |
| **Feature entropy** | **0.42%** | 50% | 75.9M | 7.87B | 5,541 |
| | 0.89% | 60% | 64.2M | 5.02B | 6,131 |

[1] "M" and "B" denote the million and billion separately.
[2] Inference speed measured on Nvidia A100 GPU.
[3] Since the pruned models are in the same architecture, their inference time are very close in practice, so it is the average value across all the models.

