# OpenReview forum: "Quantitative Performance Assessment of CNN Units via Topological Entropy Calculation"
_ICLR.cc/2022/Conference — ICLR 2022 Poster_

### Official Review · Reviewer_qJ7D · 2021-11-01

**Correctness:** 3
**Technical Novelty And Significance:** 3
**Empirical Novelty And Significance:** 3
**Recommendation:** 6
**Confidence:** 4

**Main Review:**

Strength
- The establishment of a technique for estimating the state of a neural network with respect to training will be useful for various applications, and the creation of a technique that can check the progress of training and compare trained NNs is expected to contribute to the development of AI.
- Such measurement techniques are generally plagued by scaling and randomness problems, but the entropy-based approach is a significant contribution in that it solves these problems.
- Being able to analyze the learning progress at the structural level of the NN can also provide information for ways to improve performance.
- Although it is a natural idea to measure the distribution of the activity of each unit, it is an interesting idea to construct it on a TDA basis, because a simple measurement method has many problems.
- Although the proposed method is heuristic and still in the observation stage, it is worthy of further development.

Weakness
- The authors state that the size of the unit can be assumed without loss of generality, but that seems to be a rather strong assumption. Since the proposed method can be computed only for square matrices, it cannot be applied directly when the number of elements in the data or filter is not square.
- It is also questionable that the unit is two-dimensional. In general, the middle layer of a CNN is multiple images. One of the advantages of the proposed method is that it can be measured independently of the order of the elements (i.e., the numbering of i and j). This is because the order may change in each training trial. The single-sheet assumption may not be a problem if you compare each corresponding sheet, but the order of the sheets will be switched in each trial, making it less useful.
- In TDA, we generally refer to the time of occurrence and disappearance of the simplex in filtration as birth time and death time, respectively, so calling the first non-zero state of the Betti curve the birth point may cause confusion.
- The reasons for using birth points are explained in the appendix. However, if, for example, another 1-cycle is created immediately after the first 1-cycle is created, it means that significant activity has taken place in a different region, so ignoring it would be lacking critical information. The trade-off between ease of calculation and lack of information will need to be discussed.
- Each cell of a CNN unit is fundamentally independent and does not represent any relationship. It may retain the positional relationships of the input images, but it is still counter-intuitive to represent the activity of the units in a relational graph. Therefore, it is difficult to understand why this construction method extracts the features of the activity of the unit.
- This construction method may only extract information similar to the change in the number of active cells for a change in threshold. In this case, there is no effect of using TDA. These things need to be considered and verified in order to show that the proposed method makes sense.

**Summary Of The Paper:**

In this paper, the authors propose a technique to measure the state of each neural network unit. The features of the neural network units are given by a method that applies TDA to a graph representation of the activity degree of each unit. The authors also experimentally measure the degree of learning and show its effectiveness against other methods in terms of scaling and randomness.

**Summary Of The Review:**

The effects suggested by the proposed method are very useful and can be considered a sufficient contribution. On the other hand, the proposed method is heuristic and limited, and there are some questions about the theory behind feature extraction, which require deeper investigation in the future.

[Additional comments after discussion]
Through discussion and revision, some of the unclear points were clarified and the paper became better. The proposed tool has the potential to be a very useful tool and its contribution is appreciated. On the other hand, this is something that needs to be developed in the future. These points were taken into account at the beginning, and although they are relatively positive, the score remains unchanged.

---

> ### Author Response · Authors · 2021-11-16
> **We thank the reviewer for all the constructive and valuable comments! （Part 1)**
>
> Q1. We thank the reviewer for the valuable comment, and apologize for the misdescription here. We have revised "Without loss of generality, we assume the input image for network model to be square." to "In general, input images for a network model are commonly resized to be square for processing.". We believe this is the most common case in practice, which we are focusing on. We thank the reviewer for the kindly remind.
>
> Q2.
> We thank the reviewer for this constructive comment.
>
> Firstly, we would like to explain the term unit (In footnote of the introduction). Although there is no formal definition, we follow the most commonly-used meaning of unit in [1-3]. A unit is the perceptive node in networks, which generally refers to the activated **feature map** outputted by a convolutional filter in CNNs. For a layer, there exists many units, where each unit (feature map) is 2D. It is the architecture of a CNN. For example, for VGG16, the last convolutional layer exists 512 units, and each unit has a size of $14 \times 14$.
>
> We agree with the reviewer's comment on "One of the advantages of the proposed method is that it can be measured independently of the order of the elements (i.e., the numbering of i and j)." Our method is insensitive to the change in the order of the elements.
>
> We take the 512 units in VGG for example. We would first label each unit with a specific index. For example, when we fetch the outputs of this layer, we would get all the units in an array $F$ with the size of $14 \times 14 \times 512$. We label F[:, :, 0] as Unit 0 and so on. Since no connections are between these units in architecture, so these units are independent from each other, meaning that each unit would have no effect at all to the other units at the same layer. Therefore, just as the reviewer's comment, the units could be compared between each other.
>
> During training stage, the parameters of filters could be updated by training samples, which means that the units' function may change during the updating. So, Unit 0 may not keep the same function during training. This is mainly because that the randomness introduced in the optimizer like SGD. So the indices would be shuffled during training. And if we would like to track the variation of certain attribute during training by unit' index such as Unit 0, it is meaningless. This is just the nature of optimizer, the curse of stochastic optimization. But we could track the variation by averaging the values across the layer.
>
> During testing stage, the parameters of filters are fixed. So, if we keep giving the same image to the network, it is natural that Unit 0 would not change each time, and the rest of the other units would not either. As for images in the same class, current research show that each unit would perceive certain fixed feature [1-2]. It means that this feature would be represented by the unit with this index. Researchers often present the function of unit by index. Suppose the function of Unit 0 is to perceive eyes. So, for the same class, all the images with eye feature would have representation value on Unit 0. It could not be that only part of images with eye feature are represented by Unit 0, part by Unit 4, and so on. In fact, this is how the researchers do unit interpretation [4]. If the indices are changed during testing, it would be meaningless to do this study. Moreover, since all the parameters are determined, the only stochastic factor that brings to the neural network is the input image. This would not have impact on the index of the unit since the parameters of all the filters are unchanged.
>
> We thank the reviewer for this valuable comment and we hope our response could address the concern.
>
> ```
> [1] David Bau. Network dissection: Quantifying interpretability of deep visual representations.
> [2] David Bau .Understanding the role of individual units in a deep neural network.
> [3] Ari S Morcos. On the importance of single directions for generalization.
> [4] Bolei Zhou. Revisiting the importance of individual units in cnns via ablation.
> ```
>
> Q3. We thank the reviewer for the valuable comment. We have revised "birth point" to "birth time" in the new uploaded paper. We suppose using birth time would make it clearer as the reviewer suggest.
>
> Q4. We thank the reviewer for the constructive comment. We agree on the reviewer' point that topological features could be plentiful. The rule we base on is that for effective units, their spatial pattern of active values in units between images in the same class should be similar. Moreover, it leads to those topological features should be similar.
>
> In fact, we have tried several characterizations of the Betti curve when we came up with this idea, including the maximum point of the curve and the integration during birth-death time. And after many experiments, we find that the difference would be acceptable. We have added related results in the Appendix A.3.

---

> > ### Author Response · Authors · 2021-11-16
> > **We thank the reviewer for all the constructive and valuable comments! （Part 2)**
> >
> > Q5.
> > We thank the reviewer for the valuable comment. We agree on the reviewer's point "the unit may retain the positional relationships of the input images.". We are actually on capturing the **spatial** features of the units. This spatial feature is essentially resulted from the positional relationship. Then, what is the position or spatial location in an image? In general it is considered as the indices of pixels [5-6]. So, positional relationship could be regarded as the relationship between the indices of pixels. Using graph plus topology for this analysis is straightforward and is also a quite common way.
> >
> > Units are the feature representations (feature maps) of input images. Actually, without an input, there would be no "real" unit at all. The unit itself has no stochastic property without inputs. In other words, we could not "generate" a unit without an input. It is obtained explicitly by the determined parameters in the network. The stochastic property comes from the input images.
> >
> > For different classes, this positional relationships could be different. But for images in the same class, this positional relationships would be highly regularized. So, for images in the same class, the unit may retain such class-specific relationships. In the meantime, this means that these cells in the unit are highly constrained in a spatial manner. Their indices in the unit exist certain relationship. Therefore, we map their positional relationship to a graph, which is commonly used for relational description. Essentially, we are not building the graph for the units, but for the features in the images in the same class. They are highly correlated.
> >
> > How this positional relationships would affect the graph or the following filtration? If there exists no positional relationship at all, it means that for any time in the filtration, the edge emerged in the next time would be arbitrary. In other words, any edge that has not emerged in the filtration has an equal chance to be born in the next time. In contrast, If there exists positional relationship, it means that for the filtration at current time, only part of the edges are allowed to emerge in the next time. Different positional relationship would lead to different "part of the edges" that emerges in the next time. Such behavior is actually an reflection of pixels in the natural images. For noisy images, if given some pixels, we have no idea how the rest pixels are like since any color could be possible. But for natural images, we could sense how the rest pixels should probably be like and some colors may not be emerge in the rest pixels forever.
> >
> > Therefore, the difference between no positional relationship and existing positional relationship will lead to different Betti curves. Also, different kinds of such positional relationship could also lead to distinct Betti curves. So, if we give a set of matrices with on positional relationship, their birth time should not be regularized at all, namely highly chaotic. If we give a set of matrices that share similar positional relationship, their birth time should be similar.
> >
> > We thank the reviewer for this valuable comment and hope this could address the reviewer's concern.
> >
> > ```
> > [5] Mansour, Tarek. Deep neural networks are lazy: on the inductive bias of deep learning. Diss. Massachusetts Institute of Technology, 2019.
> > [6] Mitchell, Benjamin R. The Spatial Inductive Bias of Deep Learning. Diss. Johns Hopkins University, 2017.
> > ```
> >
> > Q6.
> > We thank the reviewer for this valuable comment. We suppose this is a follow-up concern caused by the previous comment.
> >
> > We understand the reviewer implies that we do not need the emergence of "hole" in the filtration to serve as a symbolic characterization. We suppose the reviewer suggests that for one image, we could count the number of active cells in its unit for a threshold, then count the number for another threshold, and so on. Then, we could use the change of number of active cells between images in the same class for further entropy calculation. Essentially, this somehow shares similar idea with Apoz.
> >
> > Firstly, it could not satisfy our rule. Our rule is that the **spatial relationship** in the unit should be similar for images in the same class. There is no information regarding the relationship at all.
> > Secondly, it is sensitive to the scale. For example, we have two images which share the same positional relationship, but their values are several times different. In this case, they would be measured incorrectly based on counting number.
> > Thirdly, the appropriate threshold is very tricky to give. In fact, we could not give any useful threshold without a prior knowledge.
> >
> > Using topology could solve the three problems well. This is why we use topology here. Especially, we do not need to give some kinds of thresholds, and it is totally decided by the spatial nature of the units.
> >
> > We thank the reviewer for this valuable comment and hope this could address the reviewer's concern.

---

> > > ### Comment · Reviewer_qJ7D · 2021-11-18
> > > **Rebuttal clarified some of the concerns**
> > >
> > > Thank you for your clarification.
> > > Your proposal is a tool to extract information about the state of a trained NN. In the case of a tool proposal, the target data of application should be clear, so I made the point in the first review. I think A1 and A2 make those clear to some extent.
> > > From the point of view of using the tool, it is important to know what this tool can be used for and what kind of information it extracts. Your tool is heuristically created and not theoretically supported. Theoretical compensation in this area is a very difficult problem, and I think there is value in heuristic methods. However, in order to determine what it can be used for and what it is effective for, it is necessary to check various aspects. A4-A6 will be good information for this purpose. (Regarding Q5, I thought I had pointed out a possibility, but the sentence was worded in a definite way. I apologize for this.) As mentioned by other reviewers, it is important to present the analysis and use cases for this method in order to clarify the value of this tool. I hope that this will be clarified in the future.
> > > As for A3, the birth point of each cycle in filtration is sometimes called birth time. I think it is better to use the word which mean the point or time when the "first" cycle occurs.
> > > I look forward to further development of this research.

---

> > > > ### Author Response · Authors · 2021-11-19
> > > > **Thank the reviewer for the valuable comments!**
> > > >
> > > > Thank the reviewer for all the valuable comments. We are glad we have made points clearer for the reviewer. We agree with the reviewer's point on "there is value in heuristic methods". Actually, the non-linearity causes critical troubles to the root of the whole system. Many essential properties like border of the classifier is intractable. Enough empirical studies should be accumulated for the possibility of giving further ideas in theory, considering theoretical compensation in this area is very tough just as the reviewer' comment. Our paper would like to advance the accumulation of empirical studies forwards via proposing a new perspective.
> > > >
> > > > We also agree with the reviewer's point on "it is important to give use cases to clarify the value of this tool" . We suppose we have explained the function of our tool in the introduction section (paragraph 2 and 5). Then in section 4.4, we have shown the use cases of using feature entropy to discriminate 8 neural networks with different generalization in two distinct situations. What is the generalization associated with is an important problem. And also, we have gave a very comprehensive study of network pruning in Appendix A.7 including cumulative unit ablation tests and practical implementation of network pruning, even though we are targeting to make comparisons between different models while pruning focuses on the same model. We hope the two use cases could give illustrations on the uses of our tool.
> > > >
> > > > As for the terminology, in the original paper, why we use the term "birth point" here is because we do not want to confuse it with "birth time". Just as the reviewer's comment, "birth time" often refer to the birth of each cycle in filtration. Now, based on the reviewer's suggestion, we revise this with a clearer description, "Birth point is the indication of the critical element in complex filtration that begins to carry ”hole” structure (Betti number is nonzero).".
> > > >
> > > > We appreciate the reviewer for all the constructive comments.

---

### Official Review · Reviewer_CZcz · 2021-11-01

**Correctness:** 4
**Technical Novelty And Significance:** 3
**Empirical Novelty And Significance:** 2
**Recommendation:** 5
**Confidence:** 3

**Main Review:**

Strengths:
1. The paper is easy to follow and the authors clearly highlight the problems with prior approaches when measuring the importance of a unit.
2. The experimental analysis corroborates these problems nicely and shows that they can be solved by the proposed feature entropy approach.

Concerns:
1. The advantage of feature entropy over the three baseline approaches is illustrated by conducting two tests (rescaling and randomness) and observing the behaviours. However, the comparisons mostly represent a sanity-check and do not illustrate the importance and practical benefit of the proposed approach.
2. Previous approaches that measure the importance of a given node have often been compared in pruning scenarios. Is there a reason, why feature entropy can not be used in this setting?
3. The method is compared to relatively old baselines. For instance, [1] and [2]  address a similar task of measuring neutron importance (in a pruning setting).

Minor:
- It is not quite clear from the start, what is meant my “status” of a unit. Maybe “importance” of a unit will be clearer.
- Consider adding Model BA in Table 4 to improve clarity.

[1] R. Yu, A. Li, C.-F. Chen, J.-H. Lai, V. I. Morariu, X. Han, M. Gao, C.-Y. Lin, and L. S. Davis. NISP: Pruning networks using neuron importance score propagation. In CVPR, 2018.
[2] Y. He, P. Liu, Z. Wang, Z. Hu, and Y. Yang. Filter pruning via geometric median for deep convolutional neural networks acceleration. In CVPR, 2019.

**Summary Of The Paper:**

The paper proposes a topology-based approach to measure the effectiveness of a unit in a neural network for a set of inputs (i.e. images from a specific class). A graph is constructed based on a given image and the units representation and the birth point is computed. The importance of a unit is then measured as the entropy over the birth point distribution, which is estimated by sampling a set of images. Experimental results highlight the methods advantage over three baseline approaches and illustrate its ability to discriminate models with different generalisation capability.

**Summary Of The Review:**

Overall, the paper is interesting and the proposed approach of measuring unit importance is well motivated. However, as mentioned above, I do have some concerns regarding the empirical evaluation of the proposed method.

---

> ### Author Response · Authors · 2021-11-16
> **We thank the reviewer for all the constructive and valuable comments! （Part 1)**
>
> We thank the reviewer for all the valuable and constructive comments. We have performed detailed investigations for each comment. These investigations are added in the new uploaded paper: a comparisons with other related methods used in pruning (Appendix 4.6 page 16-17), and a comprehensive study of pruning when using feature entropy (Appendix 4.7 page 17-20). For brevity, we only report a concise version in the response here. We thank the reviewer for the review time.
>
> `Q1: The advantage of feature entropy over the three baseline approaches is illustrated by conducting two tests (rescaling and randomness) and observing the behaviours. However, the comparisons mostly represent a sanity-check and do not illustrate the importance and practical benefit of the proposed approach.`
>
>
> We thank the reviewer for the valuable comment. We are grateful for this comment as it points to the core idea and major advantage of our method.
>
> What we focus is to give an indicator that could behave as some kind of an intrinsic property of a unit. This indicator should conform the nature of CNNs, so it could give correct indication in various situations and further could be used in analyzing different models. This in itself is a fundamental and very challenging problem.
>
> But currently, we generally simplify this problem to give importance scores for pruning purpose (we will further discuss pruning in the next response), usually with additional practical focus like easy to compute. In pruning, the situation would be more simple and units are generally restricted under the same scale in the same model. We do not need to worry too much about whether the indication conforms the nature of CNNs or not. However, when we really face this problem between models, the situations may be far more complicated. Considering the extensive research in pruning and to our knowledge nearly no current metric could solve this problem, we suppose it is time to step forwards to really investigate this problem.
>
> Why testing rescaling and randomness are necessary is that they are the most common situations. If we want to solve this problem and to make comparisons between models, it is impossible to use a metric that can not even give correct indication before and after rescaling operation.
>
> Since this problem is fundamental, the benefit may be various. We have shown one of the most basic and natural uses of our method in Section 4.4, which gives discrimination of models with different generalization from the perspective of feature representation. What really matters to the generalization of models is also a critical problem in deep learning, and could be useful in many practical applications (such as the comment 3 by Reviewer 1). We have shown the promising that possibly tackles it from the perspective feature entropy. Additionally, we have shown implementation of network pruning via feature entropy, which is added in Appendix A.7.
>
> `Q2: Previous approaches that measure the importance of a given node have often been compared in pruning scenarios. Is there a reason, why feature entropy can not be used in this setting?`
>
> We thank the reviewer for the valuable comment. **Feature entropy could be used in pruning scenarios, but pruning scenarios could not fit our motivation.** Nonetheless, we have performed a comprehensive study of pruning which shows in Appendix, including the cumulative unit ablation tests which study the effect on the single class, and the practical implementation of network pruning.
>
> When we talk about the "importance" of a given unit and pruning, it implies two points in convention. The first point is that units are generally compared within the same model in a fixed situation. In this way, units are usually in the same scale unless some specific implementations are performed like. So it could be safe to use many conventional methods. The second point is that what really matters to network pruning is actually the importance rank between units in the single model. So, we would not bother whether the metric confirms the nature of CNN or not.
>
> As for our purpose, we have stated in the previous response. A short answer is that we are focusing on the comparisons between different models. And we show that feature entropy could also work well in the pruning implementations. Concise results are as follows (in response part 2), and the reviewer may skip the following content if checking the Appendix A.7.

---

> > ### Author Response · Authors · 2021-11-16
> > **We thank the reviewer for all the constructive and valuable comments! （Part 2)**
> >
> > **Part 1: cumulative unit ablation test.**
> >
> > For a given class, cumulative unit ablation tests check the evolution of the network performance by progressively removing each unit within a layer in a single model according to the order of certain kind of sorted attribute of units like L1-Norm, feature entropy, etc. Typically, units could be removed in two orders, one is the ascending order of the attribute, and the other one is the descending order of the attribute.
> >
> > For cumulative ablation performed via ascending order of feature entropy, since the important units have lower feature entropy, so they could be removed at first. Generally, if the accuracy drop fast, we may consider the units picked out to be removed are important to the network on the given class. So it could be used to roughly judge how well the attribute indicates the unit importance in this situation. Here, we use the number of units needed to be removed that makes the accuracy drop to chance level as the indication of this dropping speed. We compare our method with the mentioned methods in the paper. Note that we have added additional two methods used for network pruning, FPGM and NISP. We could find that feature entropy need less units to drop below chance level than other methods.
> >
> > ||L1-norm|Class selectivity|Apoz|NISP|FPGM|Feature entropy|
> > |--|:--:|:--:|:--:|:--:|:--:|:--:|
> > |Number of units need removed|71|87|141|347|400|38|
> >
> > On the contrary, for cumulative ablation performed via descending order of feature entropy, since the unimportant units have higher feature entropy, so they could be removed at first. Interestingly, we find that the testing accuracy would increase slowly from 0.54 to 1.0 when 96\% units are removed, a 0.46 performance enhancement. Therefore, here we use the accuracy enhancement value to roughly judge how well the attribute indicate the unit importance. We could find that feature entropy and class selectivity gives the most enhancement.
> >
> > ||L1-norm|Class selectivity|Apoz|NISP|FPGM|Feature entropy|
> > |--|:--:|:--:|:--:|:--:|:--:|:--:|
> > |Accuracy enhanced|0.44|0.46|0.36|0.26|0.30|0.46|
> >
> > **Part 2: Implementation of network pruning.**
> >
> > We then perform a channel-level network pruning for the whole dataset. The network is pruned by following the layer-by-layer strategy.
> >
> > In pruning, all the methods follow the same previous implementation. For short, the final results are listed in the table below. Except for the unchanged network, the accuracy reported in the table is the accuracy drop compared to the unchanged network after the models are fine-tuned.
> >
> > ||Accuracy|Pruned ratio|#Params|FLOPs|img/sec|
> > |--|:--:|:--:|:--:|:--:|:--:|
> > Unchanged network|65.71%| - | 138M |30.96B|2,178|
> > L1-Norm|1.03%| 50% | 75.9M | 7.87B |5,541|
> > Apoz|1.59%| 50% | 75.9M | 7.87B |5,541|
> > Class selectivity|0.38%| 50% | 75.9M | 7.87B |5,541|
> > FPGM|0.59%| 50% | 75.9M | 7.87B |5,541|
> > NISP|0.92%| 50% | 75.9M | 7.87B |5,541|
> > Feature entropy|0.09%| 40% | 87.8M | 11.15B |4,869|
> > Feature entropy|0.42%| 50% | 75.9M | 7.87B |5,541|
> > Feature entropy|0.89%| 60% | 64.2M | 5.02B |6,131|
> >
> > Note that img/sec is the inference speed measure on Nvidia A100 GPU. Since the pruned models are in the same architecture when pruning ratio is set to 50%, their inference time are very close in practice, so it is the average value across all the models.
> >
> > We thank again for the review time and valuable comment, and the reviewer could check Appendix for more details.
> >
> > `Q3: The method is compared to relatively old baselines. For instance, [1] and [2] address a similar task of measuring neutron importance (in a pruning setting).`
> >
> > We thank the reviewer for the valuable comment. We have compared to the two mentioned methods, and the detailed results are added in Appendix A.6.
> >
> > [1] measures the importance of a unit by propagating the final feature selection score backwards based on coefficients of network weights at a specific layer. [2] uses the norm distance to the geometric median of the set of filters as the indicator of the importance of a filter at a layer.
> >
> > It is easy to notice that the two methods are still associated with scaling of weights. So, it could not pass the rescaling tests, the results are as follows, where the same strategies are used as in Section 4.3.
> >
> > Rescaling test
> >
> > |Images|filters|Accuracy|FPGM|NISP|feature entropy|
> > |:--:|:--:|:--:|:--:|:--:|:--:|
> >  |- |- | 0.83 | 0.88 |72.37 |1.87|
> > half|-| 0.81 | 0.88 | 36.04|1.90|
> > -|half| 0.83 | 0.44 | 36.17 |1.87|
> > half|half| 0.79 |0.44 |17.81|1.92|
> >
> > `Q4: It is not quite clear from the start, what is meant my “status” of a unit. Maybe “importance” of a unit will be clearer.`
> >
> > We thank the reviewer for the valuable comment.  Why we use the word status here is that we would like to imply that feature entropy is not just an importance score but also like an intrinsic "property".
> >
> >
> > `Q5: Consider adding Model BA in Table 4 to improve clarity.`
> >
> > We thank the reviewer for the helpful advice. We have added in the new uploaded paper.

---

> ### Author Response · Authors · 2021-12-01
> **A gentle kind discussion about whether the concerns have been addressed.**
>
> We thank the reviewer for all the valuable comments. We have carefully studied all the comments and addressed them one by one. This is just a gentle kind discussion about whether the concerns have been addressed. Meanwhile, we are glad to address any further concerns. We thank the reviewer for the valuable time.

---

### Official Review · Reviewer_F2EM · 2021-11-03

**Correctness:** 3
**Technical Novelty And Significance:** 2
**Empirical Novelty And Significance:** 2
**Recommendation:** 6
**Confidence:** 4

**Main Review:**

The main essence of the paper is to translate the activations of the neural network for a given pattern class into a graph representation whose topology evolution is analyzed (using TDA) as a function of activation threshold.  The main novelty is the combination of well known methods  from topological data analysis and apply it to neural unit characterization.   The feature entropy measure defined seems to be more stable across variations do the rescaling and perturbations and the measure seems to correlate with network hyperparameter changes and perturbations. Thus the measure may be suitable for characterization of relationships between input/output in a systematic way.

However, The main weakness of the paper is that the results in the paper are not conclusive on how this work will impact neural net design choices and optimization.  If this paper can be extended to address this, I believe the work will have impact.

It is not clear to me why the authors take the activations and perform a symmetry transformation (A^v = max (A_v, (A_v)^t) ) after equation 3.   It is not clear to my why you chose 0.1 for p in equation (12).  What is the rationale here?

The experimental sections can be improved in clarity. While I follow the general results, I cannot completely comprehend the fact that the feature entropy drops rapidly and the value in the last layer is large.  Can you elaborate about this and the above points in the rebuttal?

While the ideas in the paper are clear, the writing needs to be improved.  There are several places where the sentence structure is awkward and it is hard to comprehend what the authors state.
e.g. 1)  In contrast, for ineffective units, since being incapability of effectively  ---> change to:   being incapable of effectively
2) In Hofer et al. (2017), topological signatures of data are evaluated and used to improved
the classification of shapes. --> change to: used to improve
3) rewrite line in section 3.2 -- "For a specific target class C, consider image I(i) ...  By perceived
with an ideal unit U, it should present similar pattern with other image samples."
4) section 4.2 -- when referring to figures and results the sentences are often awkward to the reader..
5) section 4.3 -- "Since random units are clearly incapable to well perceive features
like those trained units,"   should be changed to 'perceive features well'
6) section 4.3 -- reference to ApoZ seems to be wrong (As far as I understand ApoZ was proposed by Hu et al, not He et al).




**Summary Of The Paper:**

The paper applies well known methods in topological data analysis for quantitative assessment of neural net unit performance. Specifically, a scheme is proposed wherein spatial patterns of activation in the network units are determined as function of a sequence of activation thresholds, patterns transformed to a graph representations (i.e. successive filtrations) and their corresponding kth betti number, (i.e. number of kth dimensional holes), evolution calculated. In this paper k is set to 1.  For units that the authors term, effective units for a given class and a given input image class population, the analysis is expected to yield to similar birth time of the holes. Thus, a feature entropy is defined on the first birth time in the betti curves computed on a given instance and aggregating the estimated birth times over a population of instances in a given class.  The main results include the illustration of invariance of the calculated feature entropy when one does reweighting / rescaling transformations in the inputs and the illustration of correlation between the choice of hyperparameters, perturbations in data sets, and the feature entropy measure.



**Summary Of The Review:**

Overall, I think the paper is interesting and the ideas are worth pursuing. The writing needs significant improvement, the experiments are still limited.  I am not sure also not sure about the impact of the paper, in its current form, and its utility for enabling improvements in design of neural net architectures.

---

> ### Author Response · Authors · 2021-11-16
> **We thank the reviewer for all the constructive and valuable comments! （Part 1)**
>
> We thank the reviewer for all the valuable and constructive comments. We have performed detailed investigations for each comment. We have added a very comprehensive study of network pruning in Appendix A.7. For brevity, we only report a concise version in the response here. We thank the reviewer for the review time.
>
> `Q1: The main weakness of the paper is that the results in the paper are not conclusive on how this work will impact neural net design choices and optimization. If this paper can be extended to address this, I believe the work will have impact.`
>
> We thank the reviewer for this constructive comment.
>
> In this paper, we focus on giving an indicator that could behave as some kind of an intrinsic property of a unit, which could be used for further analyzing or assessing the models. We have shown one of the most basic and natural uses of our method in Section 4.4, which gives discrimination of models with different generalization from the perspective of feature representation. We find that models with better generalization capability should have lower feature entropy. We have added comparisons of ResNet models in Appendix A.5.
>
> Considering optimization, we have also gave implementation of network pruning via feature entropy, which is added in Appendix A.7 in the new uploaded paper.
>
> Note that in pruning, units are generally compared in the same model, so the scaling problem would be largely alleviated unless performing some specific operations. Although we focus on units between different models, yet we show that feature entropy could also work well in the pruning implementations. Final results are as follows, and the reviewer may skip the following content if checking the Appendix A.7.
>
> ||Accuracy|Pruned ratio|#Params|FLOPs|img/sec|
> |--|:--:|:--:|:--:|:--:|:--:|
> Unchanged network|65.71%| - | 138M |30.96B|2,178|
> L1-Norm|1.03%| 50% | 75.9M | 7.87B |5,541|
> Apoz|1.59%| 50% | 75.9M | 7.87B |5,541|
> Class selectivity|0.38%| 50% | 75.9M | 7.87B |5,541|
> FPGM|0.59%| 50% | 75.9M | 7.87B |5,541|
> NISP|0.92%| 50% | 75.9M | 7.87B |5,541|
> Feature entropy|0.09%| 40% | 87.8M | 11.15B |4,869|
> Feature entropy|0.42%| 50% | 75.9M | 7.87B |5,541|
> Feature entropy|0.89%| 60% | 64.2M | 5.02B |6,131|
>
> Note that img/sec is the inference speed measure on Nvidia A100 GPU. Since the pruned models are in the same architecture when pruning ratio is set to 50%, their inference time are very close in practice, so it is the average value across all the models.
>
> `Q2: It is not clear to me why the authors take the activations and perform a symmetry transformation (A^v = max (A_v, (A_v)^t) ) after equation 3.`
>
> We thank the reviewer for the valuable comment. It is because the adjacency matrix of an undirected graph is symmetric. This is the nature of undirected graphs. When an edge is generated in the filtration, we should appropriately describe this generation from graph to simplicial complex, so for undirected graph family, we should take this adjustment just for "correctness".

---

> > ### Author Response · Authors · 2021-11-16
> > **We thank the reviewer for all the constructive and valuable comments! （Part 2)**
> >
> >
> > `Q3: It is not clear to my why you chose 0.1 for p in equation (12). What is the rationale here?`
> >
> > We thank the reviewer for this valuable comment. First, we would like to talk about why we use Eq.12 here. Note that the feature entropy is calculated on the birth distribution that comes from the samples whose units have birth time. In practice, assume we use 100 images to build birth distribution. In common situation, we may have about 50 images that have birth point, so the birth distribution would build on the 50 images. We could safely use the normal entropy formula for calculating feature entropy. Also, we show in Appendix A.4 that under this circumstance, the feature entropy could not vary too much even if we use much more images to build birth distribution.
> >
> > But we find that when we train the network via corrupting labels with a very high corruption rate such as 80%, only several images would have birth point (the average selective rate could be only 0.03 in Fig.6). In this extreme case, this is nature because we have manually forced many images that share no common feature to be in the same class. We would not expect a single unit to be able to perceive these various features. So when we use these several images to build birth distribution, the calculated feature entropy may probably be even lower than the normal case. Apparently, it is biased here because a unit that could not perceive for almost all the images should not have very low entropy. Therefore, for consistency, we modify the normal formula to cover this extreme case, where the corresponding feature entropy is set to be the maximum.
> >
> > For the reason we choose $p = 0.1$, this is an empirical observation for sampling 100 images. This means that we would consider the calculation is valid for at least 10 images that could be perceived by the unit. We use a unit in the Model BE (trained via 80\% label corruption) as an example. The unit's selective rate is 0.03 for the 100 samples. This means that only 5 images are perceived by the unit. If we do not use the modified formula, the feature entropy is 1.09, much lower than the reference unit in normal VGG16. If we use the modified formula, it is 4.46.
> >
> > We thank the reviewer for the valuable comment and hope this could solve the reviewer's concern.
> >
> >
> >
> > `Q4: The experimental sections can be improved in clarity. While I follow the general results, I cannot completely comprehend the fact that the feature entropy drops rapidly and the value in the last layer is large. Can you elaborate about this and the above points in the rebuttal?`
> >
> >
> > We thank the reviewer for this valuable comment. We think this "the feature entropy drops rapidly and the value in the last layer is large" the reviewer refers to is the observation in Fig.6B(1), which would occur when we manually corrupt the label in the dataset. In normal cases, it would rare to see this happen.
> >
> > Firstly, the reason why this would occur is that when training corruption labels, the majority of units in the last layer present a very low selective rate while units at the other layers still present a normal selective rate. It senses like the units at the last layer are collapsed in this situation. This is because that few or nearly no common features exist between image samples in the same classes. And as demonstrated previously, the unit with a very low selective rate would be considered as having the maximum entropy in our modification. So, after modification for this abnormal case, the value of feature entropy in the last layer would be very high.
> >
> > This may not happen when we train models in convention. In common cases, the selective rate of units are generally at normal level. So, it is nature to observe the decrease of feature entropy as the layer goes deeper, where the last layer has the lowest feature entropy.
> >
> >
> > `Q5: About the sentences`
> >
> > We thank the reviewers for this help comment and apologize for this. We have revised the sentences one by one according to the reviewer's comment in the new uploaded paper. We have also carefully checked other sentences in the paper.

---

### Official Review · Reviewer_LHvg · 2021-11-08

**Correctness:** 3
**Technical Novelty And Significance:** 3
**Empirical Novelty And Significance:** Not applicable
**Recommendation:** 8
**Confidence:** 4

**Main Review:**

Strengths:
1. I enjoyed reading this paper. The proposed method is introduced clearly step-by-step with figures.
2. Main claims are well-supported. The motivation is explained in Section 1 by pointing out two flaws of previous indicators. And Section 4.3 shows how the proposed method address the flaws point-to-point through experiments.
3. The finding from Section 4.4 is interesting. The proposed indicator seems to be a good replacement for extra testing data. This would be useful in some real cases where labels are expensive and we want to use all the labeled data to train a model for the potentially best performance.

Weaknesses:
1. It would be good to repeat the experiments on ResNet. I'm curious about how skip connections affect "feature entropy".
2. Sampling 100 images seems to be too small. Any reason on the sampling size? Why not computing on all images?
3. Network pruning is mentioned in the related work. It would be interesting to see how the proposed method guide pruning.
4. Minor: It is not clear to me why the adjacency matrix needs to be symmetric. Thinking backwards, does this suggest symmetry in the feature map is important or not?
5. Minor: There is no $w_v$ in equation (3).

**Summary Of The Paper:**

This paper proposes a novel method to tell whether a unit (feature map) is effective in CNNs quantitatively, via "feature entropy". The proposed method builds a weighted graph based on a feature map and calculates the Betti curve. Then it defines the "birth point" that implies the appearance of regularized spatial pattern of notable components in the feature map. By computing the "birth point" based on a sample of images from a specified class, it can get a birth distribution, upon which the "feature entropy" is computed.

**Summary Of The Review:**

Overall, it is a solid paper with clear introductions on the proposed method and good experiment design supporting the main claims. However, the fact that experiments are all based on one kind of CNNs is concerning. I would consider it as a board-line paper with the current version.

-------------
The author feedbacks are satisfying. The improvements on the paper with answers to Q1&3 are substantial. I've raised my score.

---

> ### Author Response · Authors · 2021-11-16
> **We thank the reviewer for all the constructive and valuable comments! (Part 1)**
>
> We thank the reviewer for all the valuable and constructive comments. We have performed detailed investigations for each comment. These investigations are added in the new uploaded paper: experiments on ResNet (Appendix A.5 page 14-16), study of sample size (Appendix A.4 page 13-14), a comprehensive study of pruning when using feature entropy (Appendix 4.7 page 17-20). For brevity, we only report a concise version in the response here. We thank the reviewer for the review  time.
>
> `Q1: It would be good to repeat the experiments on ResNet. I'm curious about how skip connections affect "feature entropy".`
>
> We thank the reviewer for the constructive comment. We have performed the corresponding experiments on ResNet34 and added detailed results at Appendix A.5 (page 14-16). The results observed on ResNet are pretty similar to those on VGG16. Our investigation on ResNet follows the same line with VGG16 in the main paper.
>
> Firstly, we give rescaling and randomness comparisons between using feature entropy and other methods. Feature entropy would give a stable indication of unit status while other methods still have the mentioned flaws in these situations. This is as expected because these flaws are decided by the nature of methods not network architectures.
>
> Secondly, we perform the layer and training analysis on feature entropy. In layer analysis, we find that the feature entropy of each output layer of the residual block would increase as the layer goes deeper. This is natural since the perceived features are more representative for deep layers.
>
> |conv3_3_out|conv4_2_out|conv4_6_out|conv5_1_out|conv5_3_out|
> |:--:|:--:|:--:|:--:|:--:|
> |3.084|2.716|2.591|1.964|1.753|
>
> Interestingly, when we study the reviewer's concern on "how skip connections affect feature entropy", we find that in each residual block, feature entropy could experience a slight increase and then the "add" operation make the feature entropy decrease especially after activation. This indicates that the skip connections play an important role and would make the entropy decrease a lot.
>
> |conv5_3 conv1|conv5_3 conv2|shortcut connection|activation|
> |:--:|:--:|:--:|:--:|
> |1.894|1.975|1.912|1.753|
>
> As for training analysis, we record the feature entropy of the last convolutional block and find it also behaves very close to the training loss.
>
> |epoch 1|epoch 2|epoch 4|epoch 8|epoch 16|epoch 32|
> |:--:|:--:|:--:|:--:|:--:|:--:|
> |2.116|1.942|1.875|1.806|1.763|1.753
>
>
> Finally, we have trained another four models via partial label corruption strategy to get different generalization. Feature entropy still could effectively indicate the status of these models for providing further discrimination between these models.
>
> |Model|test acc|corrupted rate|feature entropy|
> |--|:--:|:--:|:--:|
> |Model RA|0.714|0.0|1.7893|
> |Model RB|0.382|0.2|2.4884|
> |Model RC|0.229|0.4|2.9303|
> |Model RD|0.102|0.6|3.5764|
> |Model RE|0.036|0.8|4.1022|
>
> We thank again for the valuable comment and the reviewer could check Appendix for more details.
>
> `Q2: Sampling 100 images seems to be too small. Any reason on the sampling size? Why not computing on all images?`
>
> We thank the reviewer for the valuable comments. Sampling 100 images is because we find it is an appropriate size in the trade-off between computation cost and calculation stability of feature entropy.
>
> We vary the sample size from 50 to the full size of the class (1300 images) and check its impact on feature entropy. For each sample size, we perform 100 trials of sampling. We find that for a single unit (the reference unit in the paper), the variation of feature entropy during the 100 trials is acceptable when sampling 100 images and its average value stay close to that of using full size. But when sampling 50 images, it varies intensely.
>
> ||mean(100 trails)|min|max|median|std|
> |--|:--:|:--:|:--:|:--:|:--:|
> |1300 images|1.8854|1.8854|1.8854|1.8854|0|
> |500 images|1.8885|1.85523|1.9113|1.8879|0.0097|
> |100 images|1.8899|1.8617|1.9253|1.8887|0.0142|
> |50 images|1.8995|1.8078|1.9994|1.9018|0.0410|
>
> Also, we average the feature entropy of all the units at the layer and check the impact of sample size on this averaged feature entropy. Similarly, we find that sampling 100 images could give considerable results. The bracket stands for the standard deviation over the 100 trials.
>
> |1300 images|500 images|100 images|
> |:--:|:--:|:--:|
> |2.082(0)|2.066(0.0162)|2.052(0.021)|
>
> Detailed study of the influence of sample size on feature entropy could be found in the Appendix A.4 (page 13-14).

---

> > ### Author Response · Authors · 2021-11-16
> > **We thank the reviewer for all the constructive and valuable comments! (Part 2)**
> >
> >
> > `Q3: Network pruning is mentioned in the related work. It would be interesting to see how the proposed method guide pruning.`
> >
> > We thank the reviewer for this valuable advice. We are glad to show the implementation of network pruning via feature entropy, even though we aim to give an interpretation of feature representation and investigate units between different models. In general pruning scenarios, the scaling problem would be largely alleviated unless performing some specific operations. We show that feature entropy could also work well in the pruning implementations.
> >
> > **Part 1: cumulative unit ablation test.**
> >
> > For a given class, cumulative unit ablation tests check the evolution of the network performance by progressively removing each unit within a layer in a single model according to the order of certain kind of sorted attribute of units like L1-Norm, feature entropy, etc. Typically, units could be removed in two orders, one is the ascending order of the attribute, and the other one is the descending order of the attribute.
> >
> > For cumulative ablation performed via ascending order of feature entropy, since the important units have lower feature entropy, so they could be removed at first. Generally, if the accuracy drop fast, we may consider the units picked out to be removed are important to the network on the given class. So it could be used to roughly judge how well the attribute indicates the unit importance in this situation. Here, we use the number of units needed to be removed that makes the accuracy drop to chance level. We compare our method with the mentioned methods in the paper. Note that we have added additional two methods used for network pruning, FPGM and NISP. We could find that feature entropy need less units to drop below chance level than other methods.
> >
> > ||L1-norm|Class selectivity|Apoz|NISP|FPGM|Feature entropy|
> > |--|:--:|:--:|:--:|:--:|:--:|:--:|
> > |Number of units need removed|71|87|141|347|400|38|
> >
> > On the contrary, for cumulative ablation performed via descending order of feature entropy, since the unimportant units have higher feature entropy, so they could be removed at first. Interestingly, we find that the testing accuracy would increase slowly from 0.54 to 1.0 when 96\% units are removed, a 0.46 performance enhancement. Therefore, here we use the accuracy enhancement value to roughly judge how well the attribute indicate the unit importance. We could find that feature entropy and class selectivity gives the most enhancement.
> >
> > ||L1-norm|Class selectivity|Apoz|NISP|FPGM|Feature entropy|
> > |--|:--:|:--:|:--:|:--:|:--:|:--:|
> > |Accuracy enhanced|0.44|0.46|0.36|0.26|0.30|0.46|
> >
> >
> > **Part 2: Implementation of network pruning.**
> >
> > We then perform a channel-level network pruning for the whole dataset. The network is pruned by following the layer-by-layer strategy. In pruning, all the methods follow the same practical implementation. For short, the final results are listed in the table below. Except for the unchanged network, the accuracy reported in the table is the accuracy drop compared to the unchanged network after the models are fine-tuned.
> >
> > ||Accuracy|Pruned ratio|#Params|FLOPs|img/sec|
> > |--|:--:|:--:|:--:|:--:|:--:|
> > Unchanged network|65.71%| - | 138M |30.96B|2,178|
> > L1-Norm|1.03%| 50% | 75.9M | 7.87B |5,541|
> > Apoz|1.59%| 50% | 75.9M | 7.87B |5,541|
> > Class selectivity|0.38%| 50% | 75.9M | 7.87B |5,541|
> > FPGM|0.59%| 50% | 75.9M | 7.87B |5,541|
> > NISP|0.92%| 50% | 75.9M | 7.87B |5,541|
> > Feature entropy|0.09%| 40% | 87.8M | 11.15B |4,869|
> > Feature entropy|0.42%| 50% | 75.9M | 7.87B |5,541|
> > Feature entropy|0.89%| 60% | 64.2M | 5.02B |6,131|
> >
> > Note that img/sec is the inference speed measure on Nvidia A100 GPU. Since the pruned models are in the same architecture when pruning ratio is set to 50%, their inference time are very close, so it is the average value across all the models.
> >
> > We thank again for the review time and valuable comment, and the reviewer could check Appendix for more details.
> >
> >
> > `Q4 :Minor: It is not clear to me why the adjacency matrix needs to be symmetric. Thinking backwards, does this suggest symmetry in the feature map is important or not?`
> >
> > We thank the reviewer for the valuable comment. It is because the adjacency matrix of an undirected graph is symmetric. This is the nature of undirected graphs. When an edge is generated in the filtration, we should appropriately describe this generation from graph to simplicial complex, so for undirected, we should take this adjustment just for "correctness". Actually, we would not need to force the feature map to be symmetric. What we really cares is the significant value in the active value in the unit. Such adjustment would preserve the large values. In the meantime, it would not essentially damage the spatial relationship of these values.
> >
> >
> > `Q5: Minor: There is no $w_v$ in equation (3).`
> >
> > We thank the reviewer for the helpful comment and apologize for the typo. It is actually $a^{(v)}$. We have revised it in the new uploaded paper.

---

> > > ### Comment · Reviewer_LHvg · 2021-11-22
> > > **Thank you for the feedbacks**
> > >
> > > The answers to Q1 and Q3 are very strong additions. I'd like to raise my score. Thank you for the good work.

---

> > > > ### Author Response · Authors · 2021-11-23
> > > > **Thank the reviewer for all the valuable comments.**
> > > >
> > > > We thank the reviewer for all the constructive and valuable comments! We are glad to address any further concerns.

---

### Author Response · Authors · 2021-11-16
**A description to the revised paper.**

We thank all the reviewers for their constructive and valuable comments. We have carefully investigated all the comments one by one. According to these comments, we have made many improvements to the work and have uploaded the revised paper,

Regarding the revision, we
1. fix the typos, then carefully check the sentences and make the paper more readable.
2. add a study of the sample size (Appendix A,4, page 13-14).
3. add additional results on ResNet (Appendix A,5, page 14-16).
4. add comparisons with other related methods used in pruning (Appendix A,6, page 16-17).
5. add a comprehensive study of pruning (Appendix A,7, page 17-20).

We thank the reviewer for their review time.

---

### Decision · Program_Chairs · 2022-01-20

**Decision:**

Accept (Poster)

**Comment:**

This paper proposes a new method for understanding the role and importance of individual units in convolutional neural networks. The reviewers were in agreement that the technique is novel and provides potentially valuable insights into neural network behavior. The reviewers were less certain about the utility or significance of this idea; however, the authors partially addressed this concern by adding studies of using this technique as a pruning heuristic, and future researchers will be the best judge of the paper's eventual significance. With that in mind, I recommend acceptance so that this intriguing idea can become part of the research literature and future researchers will have this opportunity.